# REArtGS: Reconstructing and Generating Articulated Objects via 3D Gaussian Splatting with Geometric and Motion Constraints

**Di Wu**[1,2] **, Liu Liu** [3]*, **Zhou Linli**[1], **Anran Huang**[3], **Liangtu Song**[1], **Qiaojun Yu**[4],
**Qi Wu**[4,5†], **Cewu Lu**[4]

1 *Hefei Institutes of Physical Science Chinese Academy of Sciences*
2 *University of Science and Technology of China*
3 *Hefei University of Technology*
4 *Shanghai Jiao Tong University*
5 *ByteDance*
Email: `wdcs@mail.ustc.edu.cn`, `liuliu@hfut.edu.cn`

## Abstract

Articulated objects, as prevalent entities in human life, their 3D representations play crucial roles across various applications. However, achieving both high-fidelity textured surface reconstruction and dynamic generation for articulated objects remains challenging for existing methods. In this paper, we present REArtGS, a novel framework that introduces additional geometric and motion constraints to 3D Gaussian primitives, enabling realistic surface reconstruction and generation for articulated objects. Specifically, given multi-view RGB images of arbitrary two states of articulated objects, we first introduce an unbiased Signed Distance Field (SDF) guidance to regularize Gaussian opacity fields, enhancing geometry constraints and improving surface reconstruction quality. Then we establish deformable fields for 3D Gaussians constrained by the kinematic structures of articulated objects, achieving unsupervised generation of surface meshes in unseen states. Extensive experiments on both synthetic and real datasets demonstrate our approach achieves high-quality textured surface reconstruction for given states, and enables high-fidelity surface generation for unseen states. Project site: `https://sites.google.com/view/reartgs/home`.

## 1 Introduction

Articulated objects are ubiquitous in our daily lives. Modeling articulated objects, i.e. mesh reconstruction and generation, holds significant importance in many fields including virtual and augmented reality [16, 36], object manipulation [40, 31], robots [33, 17] and human-object interaction [37, 12]. Currently, the surface and shape reconstruction for articulated objects is a non-trivial task due to the challenges: firstly, articulated objects exhibit complicated and diverse geometric structures with a wide range of scales. Secondly, articulated objects possess varied kinematic structures, and the static surface meshes generated from vanilla 3D reconstruction methods fail to meet practical interaction requirements. Under this circumstance, PARIS [14] attempts to use multi-view images from two states for articulated object dynamically reconstruction with neural implicit radiance fields. Nevertheless, PARIS lacks geometric constraints, leading to shape-radiance ambiguity and additional errors for motion analysis.

---

*Corresponding author
†Project leader

39th Conference on Neural Information Processing Systems (NeurIPS 2025).

In recent years, 3D Gaussian Splatting (3DGS) [10] achieves realistic and real-time novel view synthesis through explicit representation with 3D Gaussian primitives. Some subsequent works have expanded 3DGS to surface reconstruction [42, 8, 1] and dynamic reconstruction [38, 13, 4]. However, existing 3DGS-based surface reconstruction methods typically suffer from insufficient geometric constraints [42] or impose constraints by restricting the shapes of Gaussian primitives [8, 6], leading to noisy surface reconstruction results. On the other hand, current dynamic reconstruction methods via 3DGS tend to input motion time and spatial positions into neural networks to obtain the deformed positions of Gaussian primitives [38, 4].

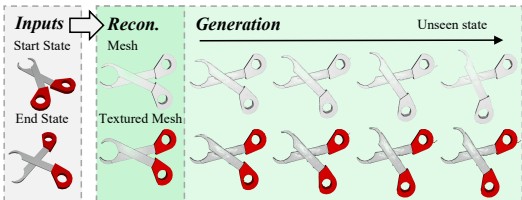

Figure 1: Given multi-view RGB images of articulated objects from two arbitrary states, our REArtGS enables high-quality textured surface mesh reconstruction and generation for unseen states.

Consequently, these methods require continuous supervision throughout the entire motion, which limits their ability to generate surface meshes in unseen states. In general, introducing 3DGS as a ready-to-use technique into articulated object surface reconstruction and generation is still not feasible and remains a challenging task.

In this paper, to address the aforementioned issues and make full advantage of 3DGS, we propose **REArtGS**, a novel approach that **R**econstructs and g**E**nerates high-quality texture-rich mesh surfaces for **Art**iculated objects via 3D **G**aussian **S**platting, only taking multi-view RGB images from two arbitrary states. Specifically, unlike most 3DGS methods based on the opacity field, which may lead to noisy surface extraction due to the non-strict linearity of the opacity field, we first introduce the Signed Distance Function (SDF) field to facilitate geometry learning. Then we propose a geometry constraint which encourages the SDF values of the Gaussian primitives approach zero when their opacity values reach the maximum , achieving unbiased guidance of the SDF field over the opacity field. In this way, we can take advantage of the strong linearity of the SDF field to explicitly establish a connection between Gaussian opacity field and the scene surface, which significantly enhances geometric learning and reduces artifacts.

Subsequently, we employ the optimized Gaussian primitives as an accurate geometry initialization for dynamic surface generation. Due to the absence of intermediate states, we leverage the kinematic structures of articulated objects to model time-continuous Gaussian deformable fields in an unsupervised fashion, and constrain the deformable fields using learnable motion parameters. Concretely, we propose a heuristic method for unsupervised part segmentation and formulate the deformable fields of dynamic parts via the motion parameters. Our REArtGS is evaluated on PartNet-Mobility [34] and AKB-48 [15] object repositories, ranging from synthetic to real-world data. Extensive experiments demonstrate that our REArtGS outperforms existing state-of-the-art methods in articulated object surface reconstruction and generation tasks, as shown in Fig. 1.

In summary, our main contributions can be summarized as follows: (1) We propose REArtGS, a novel framework introducing 3DGS to conduct high-quality textured surface reconstruction and time-continuous generation for articulated objects, only using multi-view images from two arbitrary states. (2) Our REArtGS exploits an unbiased SDF guidance for 3D Gaussian primitives to enhance geometric constraints for improving reconstruction quality, and also establishes the deformable fields constrained by kinematic structures of articulated objects to generate unseen states in an unsupervised manner. (3) We incorporate REArtGS into various scenes ranging from synthetic to the real-world data across many different articulation categories. The extensive experimental results demonstrate that our approach significantly outperforms SOTAs in both mesh reconstruction and generation tasks.

## 2 Related Work

### 2.1 Articulated Object Shape Reconstruction

Object surface reconstruction is a well-established problem for understanding the full geometric shape of objects. Some works introduce to encode continuous functions that model the objects using Signed Distance [24, 22], radiance [21, 27, 30] and occupancy [20]. To achieve both geometry and motion analysis in a single forward, Ditto [9] and REACTO [29] propose to generate shapes at unseen states from pair observations. Specifically, PARIS [14] alleviates the 3D data requirement

limitation and succeeds in reconstructing shape surface with only RGB images. Most recently, ArticulatedGS [7] and ArtGS [18] introduce 3DGS to achieve both the reconstruction and motion estimation of articulated objects. However, these methods still suffer from insufficient geometry constraints, which may lead to noisy surface mesh outputs. Therefore, our REArtGS aims to address these issues, and proposes geometric and motion constrained 3DGS for this task.

## 2.2 Surface Reconstruction with 3DGS

3D Gaussain Splatting has become increasingly popular technique for surface reconstruction in recent years [10, 11, 41, 26]. GOF [42] establishes opacity fields of 3D Gaussians using ray-tracing-based rendering, and extracts the surface meshes by the opacity level set. 2DGS [8] and PGSR [1] seek to transform 3D Gaussians into 2D flat representation, obtaining accurate normal distribution. Although improving the surface reconstruction quality, these methods still lack more reasonable geometry constraints. Several researchers attempt to integrate SDF representation with 3D Gaussians [39, 2, 35, 3], but these approaches commonly use SDF to regularize normals and guide the pruning of 3D Gaussians, without substantial optimization of Gaussian opacity fields. Meanwhile, some works conduct dynamic reconstruction using 3DGS. Deformable 3DGS proposes a deformation field to reconstruct dynamic scenes with 3D Gaussian primitives. 4DGS [4] leverages a spatial-temporal structure encoder and a multi-head Gaussian deformation decoder to derive the deformed 3D Gaussians at a given time. However, these methods rely on complete supervision of the motion process, limiting their capacity for generating unseen states. The similar issues also occur in DGMesh [13] and REACTO [29]. We provide more analysis and comparison of the related works in Appendix B.

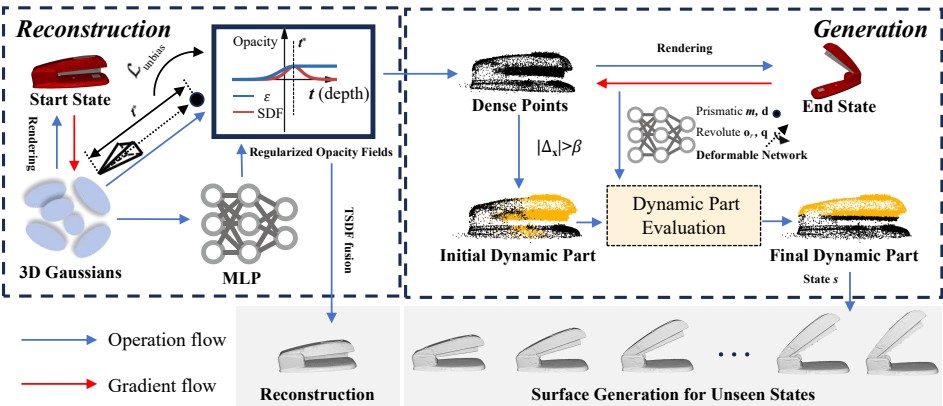

Figure 2: The overall pipeline of REArtGS. We introduce additional geometric and motion constraints for 3D Gaussian primitives, achieving high-quality surface mesh reconstruction and time-continuous generation, with only multi-view images from arbitrary two states.

## 3 Method

The overall framework of our REArtGS is illustrated in Fig. 2. Taking multi-view RGB images from two arbitrary states $s = 0$, $s = 1$ of articulated objects, we aim to achieve high-quality textured mesh reconstruction and generation at any unseen states between $s \in [0, 1]$.

We first introduce SDF representation and propose an unbiased SDF regularization to enhance the geometry constraints of 3D Gaussian primitives. In this manner, we improve the reconstruction quality and yield dense point clouds at state $s = 0$, providing an accurate geometric prior for subsequent dynamic reconstruction. Then we establish time-continuous deformable fields for 3D Gaussian primitives constrained by the kinematic structures of articulated objects. Given any state $s$, we can derive the deformed position $x_s$ of Gaussian primitives through the deformation fields in an unsupervised fashion. The details of our approach are elaborated below.

## 3.1 Reconstruction with Unbiased SDF Guidance

To better preserve the geometric features of Gaussian primitives, we use the ray-tracing-based 3DGS following GOF, which evaluates rendering contribution of Gaussian primitives directly without the 3D-to-2D projection step. However, the Gaussian primitives still suffer from insufficient geometric constraints. Concretely, the opacity fields lack an explicit link with scene surface, which may lead to noisy outputs in surface reconstruction of articulated objects.

To address the limitation, we first introduce SDF representation to guide the geometry learning of Gaussian primitives. We utilize a Multi-layer Perceptrons (MLP) with 8 hidden layers to learn the SDF values for spatial position inputs, and the scene surface can be represented by the zero-set:

$$\mathcal{S} = \left\{ \mathbf{x} \in \mathbb{R}^3 \mid f(\mathbf{x}) = 0 \right\} \tag{1}$$

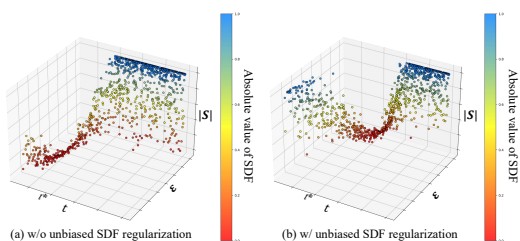

(a) w/o unbiased SDF regularization    (b) w/ unbiased SDF regularization

Figure 3: The illustration of unbiased SDF regularization. It can be observed that when $t$ approaches $t^*$, the absolute value of SDF $|\mathcal{S}|$ converges to zero with the unbiased SDF regularization.

Following Neus [32], we can derive the opacity $\hat{\boldsymbol{\sigma}}$ for the center $\mathbf{x}_i$ of Gaussian $G_i$ from learned SDF values:

$$\hat{\boldsymbol{\sigma}}_i = \max \left( \frac{\Phi\left( f\left( \mathbf{x}_i \right) \right) - \Phi\left( f\left( \mathbf{x}_{i+1} \right) \right)}{\Phi\left( f\left( \mathbf{x}_i \right) \right)}, 0 \right) \tag{2}$$

where $\Phi$ denotes a Sigmoid function. $f(\mathbf{x}_{i+1})$ is approximated as:

$$f(\mathbf{x}_{i+1}) = f(\mathbf{x}_i) + f'(\mathbf{x}_i)\Delta\mathbf{x} \tag{3}$$

where $\Delta\mathbf{x} = RS\mathbf{r}$. $R \in \mathbb{R}^{3\times3}$ is the rotation matrix derived from Gaussian's quaternion, and $S \in \mathbb{R}^{3\times3}$ is the diagonal matrix of Gaussian's scaling. $\mathbf{r}$ is the normalized ray direction. Nevertheless, the SDF-opacity conversion is not directly applicable to the $\alpha$ blending rendering. This is because each 3D Gaussian primitive models a local density distribution, while Eq. 2 only maps the global positions to their opacity values and neglects the local properties of Gaussian primitives. Therefore, we define the opacity of Gaussian primitive combining the rendering contribution $\varepsilon$ proposed in GOF as:

$$
\begin{aligned}
\boldsymbol{\sigma_i} &= \hat{\boldsymbol{\sigma}}_i \varepsilon\left( G_i \right) \\
&= \hat{\boldsymbol{\sigma}}_i \cdot \max(e^{-\frac{1}{2}\mathbf{x}_L{}^T \mathbf{x}_L}) \\
&= \hat{\boldsymbol{\sigma}}_i \cdot \max \left( e^{-\frac{1}{2}\left( \mathbf{r}_L^T \mathbf{r}_L t^2 + 2\mathbf{o}_L^T \mathbf{r}_L t + \mathbf{o}_L^T \mathbf{o}_L \right)} \right)
\end{aligned}
\tag{4}
$$

where $\mathbf{o}_L$, $\mathbf{r}_L$ are the camera center and incident ray direction represented in the local coordinate system of Gaussian $G_i$, which can be acquired by local scaling matrix and rotation matrix of $G_i$. The maximum value of $\varepsilon\left( G_i \right)$ occurs at ray depth $t^*$, expressed as follows:

$$t^* = \frac{\mathbf{o}_L^T \mathbf{r}_L}{\mathbf{r}_L^T \mathbf{r}_L} \tag{5}$$

The rendering equation based on $\boldsymbol{\sigma}$ is represented as:

$$\mathbf{C} = \sum_{i=1}^{N} \mathbf{c}_i \boldsymbol{\sigma_i} \prod_{j=1}^{i-1} \left( 1 - \boldsymbol{\sigma_i} \right) \tag{6}$$

where $N$ is the number of Gaussian primitives involved in $\alpha$-blending and $\mathbf{c}_i$ is the color modeled with spherical harmonics. To leverage SDF to constrain the geometric learning of Gaussian primitives, we first introduce a bell-shaped function $\Phi_k$ to modulate the transformation from SDF to opacity and replace the original Sigmoid activation function $\Phi$ in Eq. 2, formulated as:

$$\Phi_k(f(\mathbf{x})) = \frac{e^{k \cdot f(\mathbf{x})}}{\left( 1 + e^{k \cdot f(\mathbf{x})} \right)^2} \tag{7}$$

where $k$ is a learnable parameter that adjusts the function shape and we initially set it to 0.1. Intuitively, the Gaussian primitive closer to the surface has a higher opacity value.

However, since $\hat{\boldsymbol{\sigma}}$ is determined solely by the Gaussian's center, a non-alignment still exists between the maximum points of $\hat{\boldsymbol{\sigma}}$ and $\varepsilon$, which can be formulated as:

$$t_{\text{bias}} = \left\| \underset{t}{\operatorname{argmax}}(\hat{\boldsymbol{\sigma}}) - \underset{t}{\operatorname{argmax}}(\varepsilon) \right\| \tag{8}$$

To mitigate this bias, we propose an unbiased regularization for the SDF value at depth $t^*$ as following:

$$\mathcal{L}_{\text{unbias}} = \| f(\mathbf{o} + t^* \mathbf{r}) \|_2^2 \tag{9}$$

where $\mathbf{o}$, $\mathbf{r}$ are the camera center and incident ray direction. Through the unbiased regularization, we encourage that when the rendering contribution of a Gaussian primitive reaches its maximum value, the corresponding spatial position is close to the scene surface, as shown in Fig. 3. In this manner, the SDF representation is able to regularize the opacity fields without bias and facilitate more reasonable distribution of Gaussian primitives over scene surface. We provide more detailed elaboration and proof in the Appendix C.1.

## 3.2 Mesh Generation with Motion Constraints

Due to the explicit representation of Gaussian primitives, we can acquire accurate dense point clouds from reconstruction, serving as initial geometry for mesh generation. Then we establish time-continuous deformable fields for 3D Gaussians to infer the deformed position $\mathbf{x}_s$ at state $s$. However, it is challenging to optimize the deformable fields for Gaussian primitives only with the supervision of two states. Accordingly, we exploit the kinematic structures of articulated objects to constrain the Gaussian deformation field. We first assume that the articulated motion only includes rotation and translation following PARIS. Then we employ a heuristic method to efficiently predict the joint type through movement trend of points, which is elaborated in the Appendix C.3. For rotation joints, our learning objective is to determine their rotation axis and rotation angle. Specifically, we take the pivot point $\mathbf{o}_r \in \mathbb{R}^3$ of the rotation axis and a normalized quaternion $\mathbf{q} \in \mathbb{R}^4$ as learnable parameters, where the quaternion can be decoupled into the rotation axis $\mathbf{a} \in \mathbb{R}^3$ and rotation angle $\theta$. For prismatic joints, our learning objective is to determine their translation directions $\mathbf{d}$ and translation distances $m$. Similarly, we take the unit vector $\mathbf{d} \in \mathbb{R}^3$ representing the translation direction and the translation distance $m$ as learnable parameters.

Subsequently, we formulate the deformable fields through a canonical state. Given the rotation motion $SO(3) \in \mathbb{R}^{3 \times 3}$, we define its canonical state as $s^* = 0.5$ and the rotation angle of state $s$ can be expresses as:

$$\theta_s = \frac{(s^* - s)}{s^*} \theta \tag{10}$$

where $s \in [0, 1]$ and $\theta \in [-\pi/2, \pi/2]$. The angle-bounded parameterization is able to prevent the singularity in exponential coordinates when $\|\theta\| > \pi$. Using Rodrigues' rotation formula [25], the deformed position $\mathbf{x}_s$ can be derived as following:

$$\mathbf{x}_s = \left( \mathbf{I} + \sin(\theta_s)\mathbf{K} + (1 - \cos(\theta_s))\mathbf{K}^2 \right)(\mathbf{x} - \mathbf{o}_r) + \mathbf{o}_r \tag{11}$$

where $\mathbf{K}$ is a skew-symmetric matrix formed by the rotation rotation axis $\mathbf{a}$ and $\mathbf{I} \in \mathbb{R}^{3 \times 3}$ is a unit matrix. Meanwhile, we align the local quaternions of 3D Gaussian primitives after rotation through quaternion multiplication.

For the prismatic motion $m \cdot \mathbf{d} \in \mathbb{R}^3$, we take advantage of the vector space structure of Euclidean geometry by setting $s^* = 0$. The deformed position $\mathbf{x}_s$ can be naturally acquired by linear interpolation, formulated as:

$$\mathbf{x}_s = \mathbf{x} + s \cdot m \cdot \mathbf{d} \tag{12}$$

unlike rotation motion, this parameterization avoids singularities due to the flat Riemannian structure of $\mathbb{R}^3$.

However, the deformable fields should be applied exclusively to movable Gaussian primitives. Consequently, the Gaussian primitives need to be segmented into dynamic and static parts. To address this challenge, we propose an unsupervised method for dynamic part segmentation. We first only input the multi-view images at the end state during warm-up training iterations, and then perform an initialization segmentation of the dynamic part. The segmentation criterion for the initial dynamic

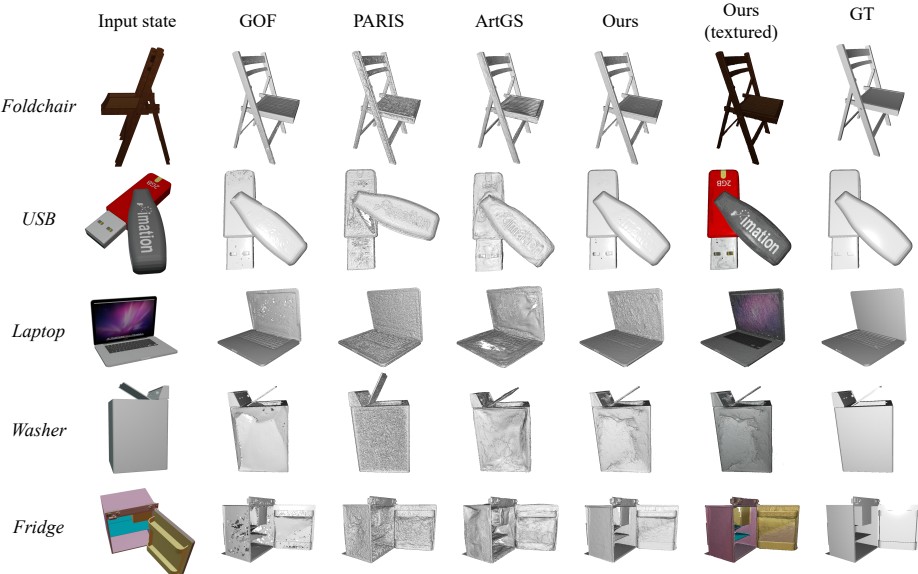

Figure 4: The qualitative result of surface reconstruction on PartNet-Mobility dataset. We show both textured and non-textured meshes for the best comparison.

components is represented as: $|\Delta_{\mathbf{x}}| > \beta$, where $|\Delta_{\mathbf{x}}|$ is the L1 distance of spatial position variation during the warm-up training, and $\beta$ is its average value. Furthermore, the spatial transformation of the dynamic part should align with the learnable motion parameters. Hence the dynamic part among the Gaussian primitives is re-evaluated at certain intervals of iterations, where the criterion for revolute or prismatic joints is represented as:

$$\left|\hat{\theta} - \theta\right| < \frac{\varphi_\theta}{K}$$
$$|\hat{m} - m| < \frac{\varphi_m}{K} \qquad (13)$$

where $\hat{\theta}$, $\hat{m}$ are the rotation angle around axis $\mathbf{a}$ and the translation distance respectively, $\varphi$ is the tolerance threshold which decreases as the number of iterations $K$ increases.

Moreover, our method can be conveniently transferred to multi-part articulated objects, through sequentially learning the segmentation masks and motions of each individual part. Please refer to our Appendix C.4 for more details.

### 3.3 Optimization and Textured Mesh Extraction

Since the shapes of Gaussian primitives are not limited, the normal distributions of Gaussian primitives are difficult to estimate accurately. However, Gaussian's normals should be parallel to the gradients of SDF in an ideal case. We consequently introduce additional regularization to regularize the normals of Gaussian primitives by SDF representation, expressed as follows:

$$\mathcal{L}_{\text{normal}} = \frac{1}{N} \sum_i^N (1 - \frac{|\mathbf{n}_i \cdot \nabla f(\mathbf{x}_i)|}{\|\mathbf{n}_i\| \cdot \|f(\mathbf{x}_i)\|}) \qquad (14)$$

where $\mathbf{n}_i$ represents the normal of Gaussian $G_i$, which is calculated following GOF:

$$\mathbf{n}_i = -R_i^T S_i^{-1} \mathbf{r}_L \qquad (15)$$

where $R$ is the rotation matrix encoded by Gaussian's quaternion and $S$ is the scaling matrix. We also employ the Eikonal regularization term [5] to encourage SDF gradient direction perpendicular to the surface, represented as:

$$\mathcal{L}_{\text{eik}} = \frac{1}{N} \sum_{i=1}^N (\|\nabla f(\mathbf{x}_i)\|_2 - 1)^2 \qquad (16)$$

The overall training objective $\mathcal{L}$ is formulated as follows:

$$\mathcal{L} = \mathcal{L}_c + \lambda_1 \mathcal{L}_{\text{unbias}} + \lambda_2 \mathcal{L}_{\text{normal}} + \lambda_3 \mathcal{L}_{\text{eik}} + \lambda_3 \mathcal{L}_d \qquad (17)$$

where $\lambda$ is the weight of regularization and $\mathcal{L}_d$ is the depth distortion loss following [8]. $\mathcal{L}_c$ is defined as:

$$\mathcal{L}_c = \lambda\mathcal{L}_{\text{D-SSIM}} + (1 - \lambda)\mathcal{L}_1 \tag{18}$$

where $\mathcal{L}_{\text{D-SSIM}}$ is proposed in [10] and $\mathcal{L}_1$ is the L1 norm of the pixel loss.

Once the optimization converges, we adopt the TSDF fusion algorithm[23] to extract the textured mesh from Gaussian primitives. We utilize the $\alpha$-blending rendering pipeline to obtain depth, opacity and RGB renderings for training views. Then we integrate these images into a voxel block grid (VBG) and extract a triangle mesh from the VBG. The textures of surface meshes can be efficiently obtained leveraging the spherical harmonics and opacity of Gaussian primitives. Please refer to our Appendix C.3 for a more detailed illustration.

Table 1: Quantitative results for the surface reconstruction quality on PartNet-Mobility dataset. We bold the best results and underline the second best results. $*$ means we implement it without depth supervision for fair comparison.

| Metrics | Method | Stapler | USB | Scissor | Fridge | Foldchair | Washer | Blade | Laptop | Oven | Storage | Mean |
|---|---|---|---|---|---|---|---|---|---|---|---|---|
| CD(ws) ↓ | A-SDF [22] | 14.19 | 7.14 | 10.61 | 13.71 | 40.85 | 12.50 | 3.31 | 2.11 | 21.37 | 22.57 | 14.84 |
| | Ditto [9] | 2.38 | 2.09 | 1.70 | 2.16 | 6.80 | **7.29** | 42.04 | 0.31 | **2.51** | **3.91** | 7.19 |
| | PARIS [14] | **0.96** | 1.80 | 0.30 | 2.68 | 0.42 | 18.31 | **0.46** | **0.25** | 6.07 | 8.12 | 3.94 |
| | GOF [42] | 1.51 | 8.09 | 0.36 | 1.51 | 0.53 | 17.35 | 0.89 | 0.84 | 19.81 | 10.04 | 6.09 |
| | ArtGS* [18] | 2.77 | 1.36 | 0.75 | 2.01 | 0.41 | 20.59 | 0.63 | 0.99 | 9.01 | 9.00 | 4.75 |
| | REArtGS (Ours) | 3.47 | **0.75** | **0.29** | **1.50** | **0.40** | 12.20 | 0.72 | 0.53 | 8.89 | 8.27 | **3.79** |
| CD(rs) ↓ | A-SDF [24] | 2.140 | 2.478 | **1.805** | 1.801 | 2.080 | 1.590 | 5.630 | 2.418 | 2.751 | 4.077 | 2.677 |
| | Ditto [9] | 2.874 | 3.049 | 2.926 | 1.550 | 0.925 | 1.428 | 5.792 | 1.184 | 1.296 | 2.837 | 2.386 |
| | PARIS [14] | 2.077 | 3.812 | 2.807 | **0.370** | 1.209 | 3.869 | 2.855 | 1.025 | 2.873 | 2.903 | 2.380 |
| | GOF [42] | 2.510 | 1.736 | 2.304 | 2.230 | 2.787 | 5.725 | 2.111 | 0.961 | 3.080 | 2.812 | 2.626 |
| | ArtGS* [18] | 2.949 | 1.704 | 2.438 | 0.725 | 0.615 | 2.874 | **2.031** | 1.182 | 1.148 | **1.157** | 1.682 |
| | REArtGS (Ours) | 2.186 | **1.433** | 2.291 | 0.475 | **0.018** | 1.204 | 2.596 | **0.038** | **0.784** | 1.330 | **1.236** |
| F1 ↑ | A-SDF [24] | 0.041 | 0.035 | 0.094 | 0.019 | 0.053 | 0.046 | 0.224 | 0.001 | 0.010 | 0.015 | 0.054 |
| | Ditto [9] | 0.197 | 0.181 | 0.275 | 0.114 | 0.352 | 0.059 | 0.107 | 0.366 | 0.052 | 0.030 | 0.173 |
| | PARIS [14] | 0.240 | 0.151 | 0.343 | 0.091 | 0.429 | 0.024 | 0.396 | **0.533** | 0.031 | 0.033 | 0.227 |
| | GOF [42] | 0.217 | 0.189 | **0.614** | **0.173** | 0.480 | **0.087** | 0.386 | 0.303 | 0.049 | 0.036 | 0.253 |
| | ArtGS* [18] | 0.251 | 0.209 | 0.438 | 0.118 | 0.447 | 0.053 | 0.408 | 0.294 | 0.056 | 0.042 | 0.232 |
| | REArtGS (Ours) | **0.256** | **0.307** | 0.598 | 0.165 | **0.502** | 0.069 | **0.488** | 0.419 | **0.065** | **0.066** | **0.294** |
| EMD ↓ | A-SDF [24] | **0.755** | 1.113 | 0.952 | 0.945 | 1.020 | 0.982 | 1.763 | 1.039 | 1.174 | 1.258 | 1.100 |
| | Ditto [9] | 1.724 | 1.308 | 1.212 | 0.619 | 0.935 | 0.841 | 1.996 | 0.417 | 0.852 | 0.970 | 1.087 |
| | PARIS [14] | 1.197 | 1.381 | **0.637** | **0.426** | 0.778 | 1.660 | 1.823 | 0.706 | 1.200 | 1.046 | 1.085 |
| | GOF [42] | 1.643 | 1.815 | 1.075 | 1.055 | 1.181 | 1.692 | 1.237 | **0.394** | 1.241 | 0.959 | 1.229 |
| | ArtGS* [18] | 1.921 | **0.677** | 0.934 | 0.637 | 0.506 | 1.154 | 1.131 | 0.502 | 1.006 | 0.787 | 0.926 |
| | REArtGS (Ours) | 1.060 | 0.847 | 1.078 | 0.485 | **0.097** | **0.777** | **1.112** | **0.111** | **0.627** | **0.755** | **0.695** |

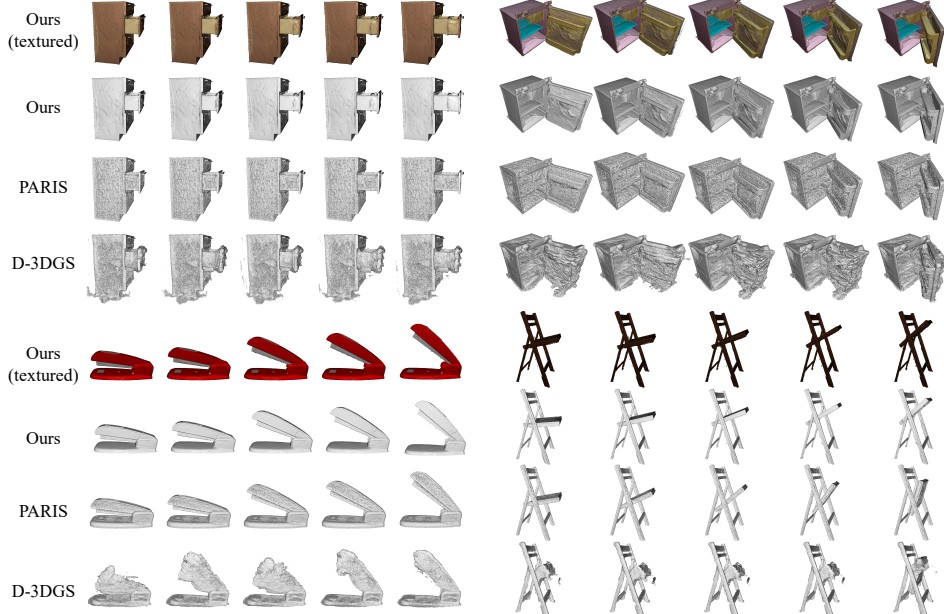

Figure 5: The qualitative results of surface generation at arbitrary unseen states on PartNet-Mobility dataset. We show both textured and non-textured meshes for best comparison. The states are sampled randomly.

Table 2: Quantitative results for the surface generation quality on PartNet-Mobility dataset. We bold the best results and underline the second best results. ∗ means we implement it without depth supervision for fair comparison.

| Metrics | Method | Stapler | USB | Scissor | Fridge | Foldchair | Washer | Blade | Laptop | Oven | Storage | Mean |
|---------|--------|---------|-----|---------|--------|-----------|--------|-------|--------|------|---------|------|
| CD(ws)↓ | A-SDF [22] | 112.30 | 34.01 | 94.11 | 33.46 | 56.73 | 55.82 | 3.78 | 27.64 | 29.60 | 25.30 | 47.28 |
| | PARIS [14] | **2.09** | 14.51 | 12.81 | 2.64 | 10.42 | 18.30 | 0.80 | 3.25 | 10.89 | **5.89** | 8.16 |
| | D-3DGS [38] | 98.90 | 16.79 | 91.84 | 48.78 | 30.10 | 30.93 | 1.262 | 65.11 | 22.39 | 18.79 | 42.49 |
| | ArtGS∗ [18] | 2.34 | 1.72 | 0.68 | **2.05** | 0.52 | 20.59 | 0.63 | **1.00** | 8.78 | 10.97 | 4.92 |
| | REArtGS (Ours) | 2.60 | **1.68** | **0.44** | 2.21 | **0.41** | 14.80 | 0.59 | 3.65 | **8.11** | 6.48 | **4.10** |
| CD(rs)↓ | A-SDF [22] | 8.992 | 1.806 | 1.822 | 5.166 | 5.668 | 7.050 | 5.991 | 2.710 | 5.245 | 7.794 | 5.224 |
| | PARIS [14] | **2.113** | **0.417** | 4.704 | **0.408** | 1.216 | **0.857** | 4.442 | 1.354 | 2.793 | 4.332 | 2.264 |
| | D-3DGS [38] | 9.589 | 1.544 | 0.891 | 0.465 | 1.147 | 3.218 | **0.168** | 0.449 | 3.773 | 3.100 | 2.434 |
| | ArtGS∗ [18] | 4.924 | 0.670 | 0.381 | 0.471 | 1.622 | 2.670 | 1.455 | 1.269 | 3.912 | **0.562** | 1.794 |
| | REArtGS (Ours) | 3.651 | 0.487 | **0.127** | 0.464 | **0.420** | 1.617 | 0.414 | **0.298** | **2.129** | 1.020 | **1.063** |
| F1↑ | A-SDF [22] | 0.007 | 0.034 | 0.056 | 0.012 | 0.021 | 0.023 | 0.168 | 0.001 | 0.010 | 0.004 | 0.034 |
| | PARIS [14] | 0.221 | 0.151 | 0.319 | 0.091 | 0.423 | 0.024 | 0.421 | 0.530 | 0.031 | 0.032 | 0.224 |
| | D-3DGS [38] | 0.064 | 0.180 | 0.125 | 0.073 | 0.168 | 0.195 | 0.233 | 0.131 | 0.022 | 0.023 | 0.121 |
| | ArtGS∗ [18] | **0.264** | **0.255** | 0.454 | 0.125 | 0.460 | 0.055 | 0.426 | 0.292 | 0.062 | 0.045 | 0.244 |
| | REArtGS (Ours) | 0.254 | 0.215 | **0.577** | **0.186** | **0.465** | 0.052 | **0.509** | **0.608** | **0.065** | **0.050** | **0.298** |
| EMD↓ | A-SDF [22] | 2.123 | 0.951 | 0.955 | 1.606 | 1.684 | 1.878 | 1.731 | 1.115 | 1.620 | 1.837 | 1.559 |
| | PARIS [14] | 2.454 | **0.457** | 1.533 | **0.451** | 0.780 | **0.655** | 1.491 | 0.770 | 1.183 | 1.354 | 1.113 |
| | D-3DGS [38] | 3.224 | 0.612 | 0.665 | 0.481 | 0.758 | 1.269 | **0.291** | 0.436 | 1.374 | 1.206 | 1.032 |
| | ArtGS∗ [18] | 2.207 | 0.562 | 0.412 | 0.563 | 0.525 | 1.171 | 0.631 | 0.724 | 1.297 | 0.750 | 0.884 |
| | REArtGS (Ours) | **1.594** | 0.494 | **0.251** | 0.473 | **0.459** | 0.900 | 0.456 | **0.353** | **1.033** | **0.689** | **0.670** |

# 4 Experiments

## 4.1 Experimental Setting

**Datasets**. To evaluate the reconstruction quality of our method, we conduct surface reconstruction experiments on synthetic dataset PartNet-Mobility [34] and real-world dataset AKB-48 [15], an extremely rich repository of real-world articulated objects which exhibit a wide variety of geometries and textures. Please see the Appendix D.1 for more details.

**Metrics**. Chamfer Distance (CD), F1-score and Earth Mover's Distance (EMD) [43] are usually adopted as evaluation metrics for surface quality. However, these methods typically focus on the whole surface, endowing the unseen areas (such as the bottom of the object) with considerable weights during evaluation. This is unreasonable since we tend to be more concerned with the upper hemisphere regions of articulated objects. Therefore, we sample point clouds from the fragments between the rays cast by test cameras and the surface mesh to calculate EMD and CD, defined as CD (rs). We also evaluate the F1-score and CD for the whole surface, defined as CD (ws), to comprehensively evaluate the surface quality. Note that the CD values are multiplied by 1000 and the distance threshold of F1-score is set to 0.4. The calculation of these metrics can be referred to in our Appendix D.4.

All the experiments are conducted on a single RTX 4090 GPU. We also provide detailed implementation of our approach and baselines in the Appendix D.2.

## 4.2 Mesh Reconstruction Performance

We report the quantitative results of mesh reconstruction in Table. 1 and qualitative results are shown in Fig. 4. Note that we implement ArtGS without depth supervision for a fair comparison. As can be observed, our REArtGS outperforms state-of-the-art approaches in the mean across all metrics with **3.79**, **1.236**, **0.294** and **0.695** respectively. Compared to A-SDF [22] and Ditto [9] that exploits 3D inputs for articulated object surface reconstruction, our method still achieves higher reconstruction quality in the majority of categories. In terms of PARIS, GOF and ArtGS, our approach exhibits significantly smoother and clearer surfaces as shown in Fig. 4, benefiting from our enhanced geometric constraints.

## 4.3 Mesh Generation Performance

We report the quantitative results of mesh generation in Table. 2 and present the qualitative results Fig. 5. Please refer to our Appendix for more qualitative results. Our REArtGS achieves the best mean results across all metrics with **4.10**, **1.063**, **0.298** and **0.670**. Compared to PARIS, our method exhibits smoother surface generation results, as shown in Fig. 5. It is mainly because PARIS lacks a reasonable geometry initiation and directly utilizes the composite rendering, leading to the spatial overlap between the neural radiation fields of the start and end states. It can be also observed that

our method is superior to ArtGS on most categories, as shown in Fig. 10 in the Appendix. This is primarily attributed to our high-quality geometry initialization from the reconstruction stage, as well as the motion-constrained deformable fields to yield dynamic generation.

Besides, we provide the part segmentation and joint parameter estimation results in the Appendix. D.5 and D.6 respectively.

Table 3: The ablation study of the unbiased SDF guidance on PartNet-Mobility dataset.

| w/ SDF | w/ Unbiased Reg. | CD (ws) | CD (rs) | F1 | EMD |
|--------|------------------|---------|---------|-------|-------|
|        |                  | 5.96    | 2.242   | 0.250 | 1.366 |
| ✓      |                  | 4.38    | 1.921   | 0.273 | 0.817 |
| ✓      | ✓                | **3.79** | **1.236** | **0.294** | **0.695** |

Table 4: The ablation study of motion constraints on PartNet-Mobility dataset.

| Settings | CD (ws) | CD (rs) | F1 | EMD |
|----------|---------|---------|-------|-------|
| w/o motion constraints | 18.65 | 2.270 | 0.209 | 1.105 |
| w/ motion constraints | **5.41** | **1.063** | **0.276** | **0.670** |

## 4.4 Ablation Studies

We conduct an ablation of the unbiased SDF guidance on PartNet-Mobility dataset to validate its effectiveness. The ablation setting and results are reported in Table. 3. The quantitative results prove that both SDF guidance and the unbiased regularization of SDF indeed improve surface reconstruction quality. This is primarily because the SDF representation establishes explicit geometric association between the opacity fields and scene surface through $\Phi_k$, and the unbiased SDF regularization further optimizes the distribution of Gaussian opacity fields.

We also conduct an ablation study of the motion constraints on PartNet-Mobility dataset and present the quantitative results in Table. 4. For the baseline without mo-

Table 5: The quantitative results on real-world AKB-48 dataset. We bold the best results and underline the second best results. ∗ means we implement it without depth supervision for fair comparison.

| Category | Method | Reconstruction | | | | Generation | | | |
|----------|--------|--------|--------|------|------|--------|--------|------|------|
|          |        | CD(ws)↓ | CD(rs)↓ | F1↑ | EMD↓ | CD(ws)↓ | CD(rs)↓ | F1↑ | EMD↓ |
| Box | PARIS [14] | 4.69 | 1.651 | 0.097 | 0.843 | 3.98 | 2.131 | 0.306 | 1.055 |
|     | ArtGS* [18] | 8.61 | 0.967 | 0.143 | 0.697 | 10.23 | 1.606 | 0.309 | 0.851 |
|     | REArtGS (Ours) | **2.49** | **0.898** | **0.205** | **0.671** | **1.28** | **1.425** | **0.559** | **0.845** |
| Stapler | PARIS [14] | **0.23** | 0.104 | 0.551 | 0.165 | 0.47 | 0.198 | 0.540 | 0.194 |
|         | ArtGS* [18] | 51.04 | 1.364 | 0.476 | 0.827 | 5.13 | 0.527 | 0.533 | 0.514 |
|         | REArtGS (Ours) | 0.24 | **0.037** | **0.728** | **0.137** | **0.34** | **0.044** | **0.582** | **0.151** |
| Scissor | PARIS [14] | 0.18 | 0.302 | 0.710 | 0.314 | 0.18 | 0.329 | 0.424 | 0.310 |
|         | ArtGS* [18] | 11.43 | 0.073 | 0.588 | 0.192 | 5.53 | 0.089 | 0.600 | 0.213 |
|         | REArtGS (Ours) | **0.08** | **0.017** | **0.902** | **0.093** | **0.08** | **0.003** | **0.899** | **0.042** |
| Cutter | PARIS [14] | 2.56 | 1.371 | 0.570 | 0.302 | 45.02 | 1.684 | 0.309 | 1.183 |
|        | ArtGS* [18] | 5.65 | 0.453 | 0.644 | 0.477 | 65.64 | **1.298** | 0.093 | 0.806 |
|        | REArtGS (Ours) | **0.13** | **0.016** | **0.874** | **0.090** | 51.56 | 1.559 | 0.228 | 0.884 |
| Drawer | PARIS [14] | 30.63 | 4.568 | 0.062 | 1.577 | 33.94 | 3.865 | 0.091 | 1.430 |
|        | ArtGS* [18] | 19.41 | 7.478 | 0.086 | 1.935 | 12.60 | **1.624** | 0.094 | **0.911** |
|        | REArtGS (Ours) | **10.73** | **2.607** | **0.138** | **1.142** | **11.71** | 1.739 | **0.209** | 0.933 |
| Eyeglasses | PARIS [14] | 0.16 | 1.582 | 0.637 | 0.212 | 0.18 | 1.550 | 0.609 | 1.284 |
|            | ArtGS* [18] | 27.49 | 1.384 | 0.557 | 0.832 | 12.12 | 1.592 | 0.747 | 0.955 |
|            | REArtGS (Ours) | **0.04** | **0.018** | **0.969** | **0.096** | **0.04** | 1.568 | **0.974** | **0.886** |
| Mean | PARIS [14] | 6.41 | 1.336 | 0.438 | 0.569 | 13.96 | 1.626 | 0.380 | 0.707 |
|      | ArtGS* [18] | 20.61 | 1.953 | 0.416 | 0.826 | 18.54 | 1.123 | 0.396 | 0.708 |
|      | REArtGS (Ours) | **2.86** | **0.599** | **0.636** | **0.372** | **10.83** | **1.056** | **0.575** | **0.624** |

tion constraints, we employ three MLPs to learn the variations of spatial positions, scales, and quaternions respectively for Gaussian primitives, which is similar to D-3DGS. The significantly superior results demonstrate the effectiveness of the motion constraints.

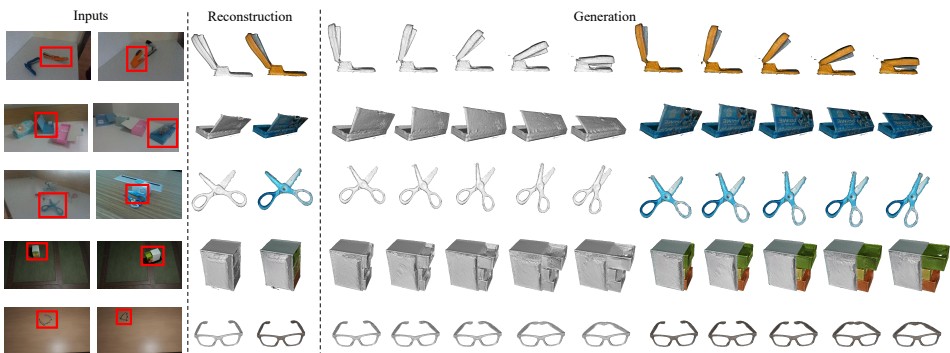

Figure 6: The qualitative results of surface reconstruction on real-world AKB-48 [15] repository. We show both textured and non-textured meshes for two-part and multi-part articulated objects.

## 4.5 Generalization to the Real World

We conduct reconstruction and generation experiments in the real world to investigate the generalization capacity of our REArtGS. We report the quantitative results comparing with PARIS and ArtGS in Table. 5, and present the qualitative results in Fig. 6. Our approach significantly outperforms PARIS and ArtGS in mean performance across all metrics. This demonstrates that our method exhibits strong generalization capability for real-world objects and enables high-quality surface reconstruction and generation as shown in Fig. 6.

# 5 Conclusion and Future Work

We propose REArtGS, a novel framework that achieves high-quality textured mesh reconstruction and dynamic generation of articulated objects using only RGB views of two arbitrary states. We propose an unbiased SDF guidance to regularize Gaussian opacity fields and model the Gaussian deformation fields constrained by kinematic structures for generating meshes of unseen states. Experiments show the superior performance of our method on both synthetic and real-world data.

The limitations of REArtGS lie on the requirement of camera pose prior and challenges of objects with transparent materials. Future works will introduce relative pose estimation to alleviate the dependence on camera poses and employ physically-based networks to model the transparent materials.

## Acknowledgements

This work was supported by National Natural Science Foundation of China under Grant 62302143, 62576130, and Anhui Province's Key Science and Technology Project 202423k09020007.

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

# Appendix

## A  Overview

In the appendix, we first provide a detailed comparison highlighting our advantages over related works in Sec. B. Then we elaborately extend our method in Sec. C from the following four aspects: unbiased SDF guidance theory, textured mesh extraction, joint type prediction, and multi-joint mesh generation. In addition, more experimental details and results are presented in Sec.D. Finally, we provide our codes and video in Sec. E.

## B  Related Work, Extended

Recently several methods also attempt to integrate SDF representaion with 3D Gaussian primitives [39, 2, 35, 3, 19] and have improved surface reconstruction quality. However, most approaches are limited to normal regularization and pruning strategy of Gaussian primitives through SDF representation [39, 2, 35, 3], and the opacity of Gaussian primitives holds a more essential role in the $\alpha$-blending. In contrast, we leverage the unbiased SDF guidance to regularize Gaussian opacity fields, achieving substantial geometric constraints and enhancing geometric optimization. 3DGSR [19] directly obtains the opacity of Gaussian primitives through SDF-opacity conversion proposed in Neus [32] and also employs a bell-shaped activation function to control the distribution of SDF-opacity. Nevertheless, since the SDF network simply utilizes Gaussian's center as input, 3DGSR ignores the local features of Gaussian primitives. In comparison, our approach combines the local rendering contributions with SDF-opacity conversation results as Gaussian's opacity values, yielding more reasonable distribution of Gaussian opacity fields.

Most recently, ArtGS [18] also proposes a mesh reconstruction method for articulated objects. Although demonstrating its effectiveness and robustness for mesh reconstruction, it relies on depth inputs, limiting its performance when the 3D supervision is unavailable. Moreover, it still lacks sufficient geometry constraints, leading to noisy results occasionally. In contrast, our REArtGS only requires 2D RGB images as inputs and introduces additional geometry constraints, improving the mesh reconstruction quality.

## C  Methodology, Extended

### C.1  Unbiased SDF Guidance for Gaussian Opacity Fields, Extended

As defined in the main draft, the opacity of Gaussian primitives combines with the rendering contribution $\varepsilon$, formulated as:

$$
\begin{aligned}
\boldsymbol{\sigma_i} &= \hat{\boldsymbol{\sigma}}_i \varepsilon \\
&= \hat{\boldsymbol{\sigma}}_i \cdot \max \left( e^{-\frac{1}{2}\left( \mathbf{r}_L^T \mathbf{r}_L t^2 + 2\mathbf{o}_L^T \mathbf{r}_L t + \mathbf{o}_L^T \mathbf{o}_L \right)} \right)
\end{aligned}
\tag{19}
$$

where $\mathbf{x}_L = \mathbf{o}_L + t\mathbf{r}_L$. $\mathbf{x}_L, \mathbf{o}_L, \mathbf{r}_L$ represent the spatial position, camera center, and ray direction of Gaussian's local coordinate system respectively. $\varepsilon$ attain its maximum value when $t = t^*$ as mentioned in the main draft. $\hat{\boldsymbol{\sigma}}_i$ can be derived from $\Phi_k$ and the SDF-opacity conversation, where $\Phi_k$ is also defined in the main draft, formulated as:

$$
\Phi_k(f(\mathbf{x})) = \frac{e^{k \cdot f(\mathbf{x})}}{\left( 1 + e^{k \cdot f(\mathbf{x})} \right)^2}
\tag{20}
$$

Although the composite function $\hat{\boldsymbol{\sigma}}\left( f(\mathbf{x}) \right)$ attains its maximum when $f(\mathbf{x}) = 0$, a non-alignment still exists between the maximum points, formulated as:

$$
\hat{t} = \operatorname{argmax}\left( \hat{\boldsymbol{\sigma}}\left( \mathbf{x}\left( t \right) \right) \right) \not\Rightarrow \hat{t} = t^*
\tag{21}
$$

This mismatch leads to a spatial deviation between $\hat{\boldsymbol{\sigma}}$ and $\varepsilon$ proposed in the main draft:

$$
t_{\text{bias}} = \left\| \operatorname*{argmax}_{t}(\hat{\boldsymbol{\sigma}}) - \operatorname*{argmax}_{t}(\varepsilon) \right\|
\tag{22}
$$

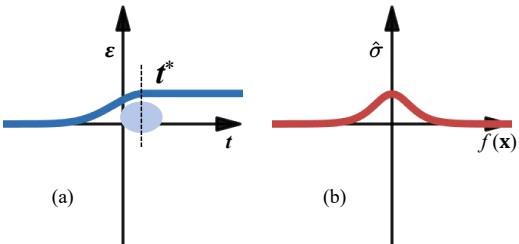

Figure 7: The distribution of $t$-$\varepsilon$ (a) and SDF-$\hat{\sigma}_i$ (b) without the unbiased SDF regularization.

As illustrated in the main draft (Fig. 3), the deviation $t_{bias}$ results in a disordered distribution of the Gaussian opacity fields. We also present their distributions without regularization for intuitive illustration in Fig. 7.

To tackle the deviation, we propose an unbiased SDF regularization $\mathcal{L}_{\text{unbias}}$ defined in the main draft:

$$\mathcal{L}_{\text{unbias}} = \|f(\mathbf{o} + t^*\mathbf{r})\|_2^2 \tag{23}$$

We present the theorem and proof that $\mathcal{L}_{\text{unbias}}$ is capable of eliminating the deviation $t_{\text{bias}}$ in Theorem. 1 and Proof. 1 below.

**Theorem 1** *Let $f : \mathbb{R}^3 \to \mathbb{R}$ be a multilayer perceptron (MLP) with smooth activation functions, and let the spatial position be defined as $\mathbf{x}(t) = \mathbf{o} + t\mathbf{r}$ with camera center $\mathbf{o} \in \mathbb{R}^3$ and ray direction $\mathbf{r} \in \mathbb{R}^3$. Let $\underset{t}{argmax}(\varepsilon) = t^*$. Consider the activation function $\Phi_k : \mathbb{R} \to \mathbb{R}$ defined by:*

$$\Phi_k(z) = \frac{e^{kz}}{(1 + e^{kz})^2}, \quad k > 0 \tag{24}$$

*If $\lim\limits_{t \to t^*} f(\mathbf{x}(t)) = 0$, then the composite function $\Phi_k \circ f \circ \mathbf{x}(t)$ attains its maximum at $t = t^*$, i.e.,*
$$\left\| \underset{t}{argmax}(\hat{\boldsymbol{\sigma}}) - \underset{t}{argmax}(\varepsilon) \right\| = 0.$$

**Proof 1** *Define $z(t) := f(\mathbf{x}(t))$. Compute the first derivative:*

$$\begin{aligned}
\frac{d\Phi_k}{dz} &= \frac{d}{dz}\left(\frac{e^{kz}}{(1 + e^{kz})^2}\right) \\
&= \frac{ke^{kz}(1 + e^{kz})^2 - 2ke^{kz}e^{kz}(1 + e^{kz})}{(1 + e^{kz})^4} \\
&= \frac{ke^{kz}(1 - e^{kz})}{(1 + e^{kz})^3}
\end{aligned}$$

*Setting $\frac{d\Phi_k}{dz} = 0$ gives the unique critical point at $z = 0$.*

*Evaluate the second derivative at $z = 0$:*

$$\begin{aligned}
\frac{d^2\Phi_k}{dz^2}\bigg|_{z=0} &= \frac{k^2 e^{kz}(1 - 4e^{kz} + e^{2kz})}{(1 + e^{kz})^4}\bigg|_{z=0} \\
&= \frac{k^2(-4)}{16} = -\frac{k^2}{4} < 0
\end{aligned}$$

*For $z > 0$, $\frac{d\Phi_k}{dz} < 0$; for $z < 0$, $\frac{d\Phi_k}{dz} > 0$. This proves that the maximum value of $\Phi_k$ is attained when $z(t) = 0$. Given that $\lim\limits_{t \to t^*} f(\mathbf{x}(t)) = 0$ and $\underset{t}{argmax}(\varepsilon) = t^*$, it can be proved that*

$$\left\| \underset{t}{argmax}(\hat{\boldsymbol{\sigma}}) - \underset{t}{argmax}(\varepsilon) \right\| = 0.$$

## C.2 Textured Mesh Extraction, Extended

In this paper, we employ TSDF fusion [23] to extract textured meshes from the regularized Gaussian opacity fields. We first obtain multi-view RGB renderings through $\alpha$-blending, where the opacity $\sigma$ and view-dependent spherical harmonics (SH) coefficients jointly modulate the color accumulation. These per-pixel blended colors are then projected into the voxel block grid (VBG), ensuring that high-opacity regions dominate the color field, while low-opacity areas are suppressed. Meanwhile, we acquire the multi-view depth renderings via $\alpha$-blending and conduct filtration with a minimum opacity threshold 0.5 as well as a maximum depth threshold 5.0. Then the filtered depth renderings are similarly integrated into the VBG. After multi-view fusion, a triangle mesh is extracted via Marching Cubes and vertex colors are directly sampled from the VBG's color fields. This process effectively bakes the fused texture into the extracted mesh.

In the early experiments, we also attempt to use SDF combined with Marching Tetrahedra to extract surface meshes, but this leads to a decline in surface quality. It is mainly because the SDF-opacity conversion only uses the center positions of Gaussian primitives for approximate calculation during training. Therefore, the SDF network only learns the spatial information at Gaussian's center, making it challenging to handle dense sampling points in mesh extraction. In contrast, the $\alpha$-blending in TSDF fusion incorporates the local information of the Gaussian primitives via Gaussian opacity fields, yielding higher-quality mesh results.

## C.3 Joint Type Prediction

Given an articulated object with unknown kinematic structure, we first obtain two coarse point clouds $\mathbf{P}_0$ and $\mathbf{P}_1$ using the multi-view images from two states $s_0$ and $s_1$ respectively, based on the proposed pipeline in Sec. 3.1. Note that it only takes few minutes since the training process only undergoes 3000 iterations. Then we use the Farthest Point Sampling (FPS) to produce the sparse point clouds $\mathbf{P}_0^*$ and $\mathbf{P}_1^*$ for $\mathbf{P}_0$ and $\mathbf{P}_1$. For each point $p_i \in \mathbf{P}_0^*$, we identify a point $q_i \in \mathbf{P}_1^*$ closest to it. Subsequently, we filter out the dynamic points $\mathbf{D}$ with significant displacement through the distance between $p$ and $q$, formulated as:

$$\mathbf{D} = \{p_i | \, \|p_i - q_i\| > \tau\} \tag{25}$$

where $\tau$ is the dynamic threshold.

After that, we determine the motion type of the dynamic point set $\mathbf{D}$ through the variance $\sigma_{\mathbf{D}}$ of the displacement direction angle.

$$\sigma_{\mathbf{D}} = \frac{1}{|\mathbf{D}|} \sum_i \left\| \arccos \left( \frac{v_i \cdot v_{\text{mean}}}{\|v_i\| \, \|v_{\text{mean}}\|} \right) \right\|^2 \tag{26}$$

where $v_i = p_i - q_i$ and $v_{\text{mean}} = \frac{1}{|\mathbf{D}|} \sum v_i$. Then we use a threshold $\sigma_{rot}$ to predict the motion type, that is, the joint type of the articulated object. Concretely, we determine the dynamic part as prismatic joint if $\sigma_{\mathbf{D}} \leq \sigma_{rot}$ and as rotation joint in unsatisfactory cases.

## C.4 Mesh Generation of Multi-Part Articulated Object

Mesh generation of multi-part articulated objects can be treated as a straightforward extension of the two-part task for our approach. We can generate meshes for multi-part objects in unseen states through a sequentially learning strategy. Specifically, given an articulated object with $k$ dynamic parts $\omega_{d_k}$, we start with fixing parts $\{\omega_{d_2}, ..., \omega_{d_k}\}$ except for the dynamic part $\omega_{d_1}$, and the object can be decomposed the into dynamic part $\omega_{d_1}$ and static part $\omega_s$ (all the remaining points). After learning the motion and segmentation mask for $\omega_{d_1}$, we sequentially perceive dynamic part $\omega_{d_2}$, while the static part $\omega_s$ can be updated as: $\omega_s = \hat{\omega}_{d_2} - \omega_{d_1}$, where $\hat{\omega}_{d_2}$ denotes the remaining points except $\omega_{d_2}$. By parity of reasoning, we achieve learning all motions and segmentation masks for the $k$ parts, and the mesh generation results for any unseen state can be obtained using the proposed pipeline in Sec. 3.2.

# D Experiments, Extended

## D.1 Dataset

The PartNet-Mobility dataset [34] is a synthetic dataset comprising articulated objects across 46 categories, characterized by a diverse scale and geometry. Following PARIS, we selected the same 10 categories for our experiments. Each category comprises 64 to 100 RGB views at two arbitrary states, which are sampled randomly from their upper hemispheres. AKB-48 [15] is a real-world dataset containing an extensive collection of articulated objects scanned in real indoor scenes. We select 6 categories with both two-part and multi-part objects and sample 100 RGB views at two arbitrary states for each object.

## D.2 Implementation

All the experiments are conducted on a single RTX 4090 GPU. Our approach constitutes a two-stage procedure to perform surface reconstruction and generation respectively. For the reconstruction stage, we train the 3D Gaussian primitives with SDF network for 3,1000 iterations, which includes an warm-up phase consisting of 1,000 iterations for SDF initial learning. Then we employ the unbiased SDF regularization in 1,500 to 1,6000 iterations, use the SDF-normal regularization and Eikonal regularization during the entire training process. We perform the densification and pruning for 3D Gaussian primitives from 1,500 iterations to 16,000 iterations. The training time of reconstruction stage is about 60 minutes.

For the generation stage, we leverage the 3D Gaussian primitives optimized from the reconstruction stage as initialization. We also set a warm-up phase with 3,000 iterations to train 3D Gaussian primitives only using RGB images from the end state. Then we start to evaluate dynamic parts of the articulated object and apply the deformation fields to them. The total number of training iterations during the generation stage is 30,000. The training time of the generation stage is about 10 minutes and the total training time for surface generation from scratch is approximately 70 minutes.

## D.3 Baselines

For the surface reconstruction experiments of articulated objects, we define the start states of articulated objects as reconstruction targets, and compare the reconstruction results with existing state-of-the-art methods: A-SDF [22], Ditto [28], PARIS [14], GOF [42] and ArtGS [18]. Note that we implement ArtGS without depth supervision for a fair comparison. In particular, A-SDF uses sampled point clouds as inputs and Ditto requires both point clouds and united robotics description format (URDF) files of articulated objects. Following PARIS, we train A-SDF on the 10 categories of the PartNet-Mobility dataset and employ the pretrained weights of Ditto to conduct the comparison.

As for the surface generation experiments, we compare the surface generation results with existing state-of-the-art methods: A-SDF, PARIS, Deformable-3DGS (D-3DGS) [38] and ArtGS [18]. Since the ground truth point clouds of unseen states are unavailable, we take the end states of the articulated objects as generation targets and generate surface meshes from the canonical states. Besides, all the methods employ point clouds (A-SDF) or RGB images (PARIS, D-3DGS, ArtGS, ours) from both start and end states as training inputs.

## D.4 Metrics, Extended

We employ CD (ws), CD (rs), F1-score, EMD as the metrics of surface quality. We present the detailed calculation below.

The CD (ws) is defined as the mean of the shortest distance between two point clouds, which can be formulated as following:

$$
\begin{aligned}
\text{CD (ws)} = \frac{1}{2} \Big( \frac{1}{N_1} \sum_{\mathbf{x} \in N_1} \min_{\mathbf{y} \in N_2} \|\mathbf{x} - \mathbf{y}\|_2^2 \\
+ \frac{1}{N_2} \sum_{\mathbf{y} \in N_2} \min_{\mathbf{x} \in N_1} \|\mathbf{y} - \mathbf{x}\|_2^2 \Big)
\end{aligned}
\tag{27}
$$

where $N_1$, $N_2$ represent the point number of output and ground truth respectively. The CD (rs) is calculated by the sampled point clouds from the fragments between the rays cast by test cameras and the surface mesh, shown as following:

$$\text{CD (rs)} = \frac{1}{2}\Big(\frac{1}{N_{s_1}} \sum_{\mathbf{x}\in N_{s_1}} \min_{\mathbf{y}\in N_{s_2}} \|\mathbf{x}-\mathbf{y}\|_2^2$$
$$+\frac{1}{N_{s_2}} \sum_{\mathbf{y}\in N_{s_2}} \min_{\mathbf{x}\in N_{s_1}} \|\mathbf{y}-\mathbf{x}\|_2^2\Big) \tag{28}$$

where $N_{s_1}$, $N_{s_2}$ denote the point number of sampled output and sampled ground truth respectively. Similarly, we adopt the same point cloud sampling method to calculate EMD score [43], which can be formulated as:

$$\text{EMD}\,(N_{s_1}, N_{s_2}) = \min_{\phi:N_{s_1}\to N_{s_2}} \frac{1}{|N_{s_1}|} \sum_{x\in N_{s_1}} \|\mathbf{x}-\phi(\mathbf{x})\| \tag{29}$$

where $\phi$ is a bijection to identify the optimal correspondence between two point clouds.

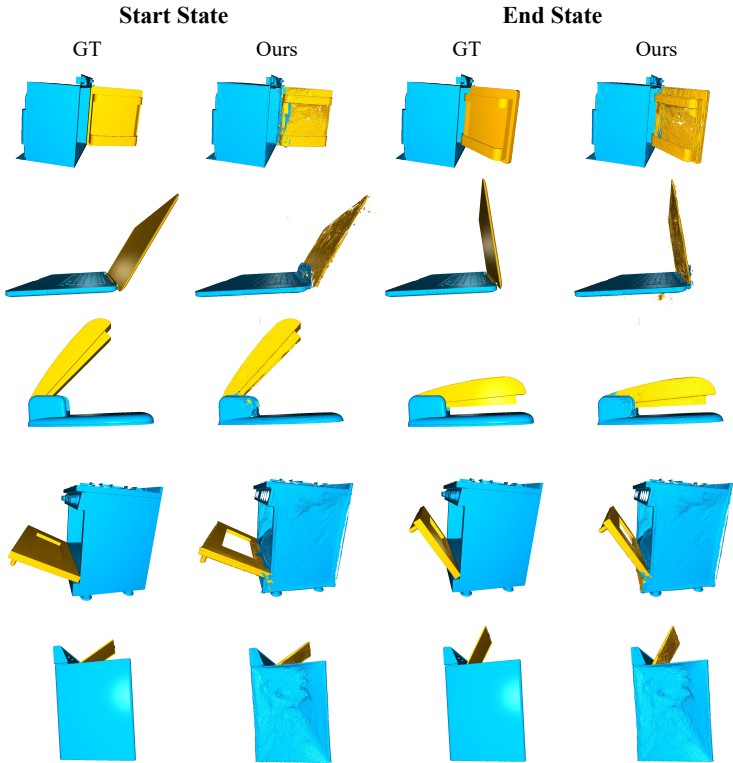

Figure 8: The qualitative results of part segmentation. We set the dynamic parts to yellow and the static parts to blue.

F1-score is the harmonic mean between the precision $P(d)$ and recall $R(d)$ with distance threshold $d = 0.4$ in this paper, formulated as:

$$F_1 = \frac{2P(d)R(d)}{P(d)+R(d)} \tag{30}$$

### D.5 Part Segmentation

We present the quantitative results of part reconstruction quality on PartNet-Mobility dataset in Table. 6, and provide the qualitative results of part segmentation in Fig. 8. Our method achieves the best mean results on both dynamic and static parts. It can be also observed that our REArtGS

successfully acquires accurate part decompositions, as shown in Fig. 8. This demonstrates that our approach is capable of perceiving motion, and enables accurate part segmentation using the simple yet effective heuristic method.

Table 6: Quantitative results for the part reconstruction quality on PartNet-Mobility dataset. CD-s ↓ and CD-m ↓ represent the CD results of static part and dynamic part respectively. We bold the best results and underline the second best results.

| Metrics | Method | Stapler | USB | Scissor | Fridge | Foldchair | Washer | Blade | Laptop | Oven | Storage | Mean |
|---|---|---|---|---|---|---|---|---|---|---|---|---|
| CD-s ↓ | Ditto [9] | 2.38 | 2.09 | 1.70 | 2.16 | 6.80 | 7.29 | 42.04 | 0.31 | 2.51 | 3.91 | 7.19 |
| | PARIS [14] | 0.96 | 1.80 | 0.30 | 2.68 | 0.42 | 18.31 | 0.46 | 0.25 | 6.07 | 8.12 | 3.94 |
| | ArtGS [18] | 3.45 | 2.05 | 0.57 | 2.03 | 0.16 | 20.97 | 0.44 | 0.76 | 9.20 | 13.51 | 5.31 |
| | REArtGS (Ours) | 1.47 | 0.82 | 0.25 | 1.53 | 0.31 | 13.04 | 0.69 | 0.48 | 6.59 | 7.30 | 3.25 |
| CD-m ↓ | Ditto [9] | 31.21 | 15.88 | 20.68 | 0.99 | 141.11 | 12.89 | 195.93 | 0.19 | 0.94 | 2.20 | 42.20 |
| | PARIS [14] | 0.85 | 0.89 | 0.23 | 1.13 | 0.53 | 0.27 | 5.13 | 0.14 | 0.43 | 20.67 | 3.06 |
| | ArtGS [18] | 2.78 | 1.17 | 0.67 | 1.43 | 0.57 | 2.82 | 1.80 | 0.99 | 2.12 | 7.18 | 2.15 |
| | REArtGS (Ours) | 1.18 | 0.40 | 0.21 | 0.66 | 0.30 | 0.42 | 1.97 | 0.11 | 0.25 | 12.39 | 1.79 |

## D.6 Joint Parameter Estimation

We also present the joint parameter estimation quantitative results in Table. 7 and qualitative results in Fig. 9 to evaluate motion learning. Following PARIS [14], we compute the orientation error (Ang Err) between the predicted axis direction ($a$, $d$ in our method) and the ground truth, and evaluate the position error (Pos Err) for revolute joints through the position difference of the pivot point ($o_r$ in our method). It can be observed that our method achieves the best results, which demonstrates our superiority in motion learning and further demonstrates the high fidelity of our dynamic generation.

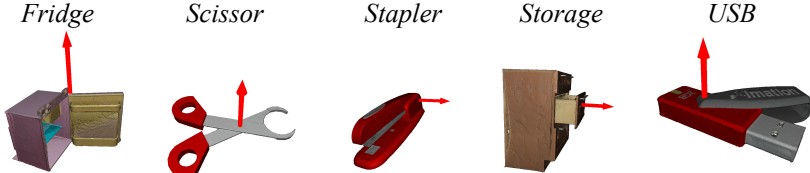

*Fridge*      *Scissor*      *Stapler*      *Storage*      *USB*

Figure 9: The qualitative results of joint parameter estimation. We show the textured mesh with red arrow representing joint axis.

Table 7: Quantitative results for the joint parameter estimation on PartNet-Mobility dataset. Note that, - denotes the absence results for prismatic joints. We bold the best results and underline the second best results.

| Metrics | Method | Stapler | USB | Scissor | Fridge | Foldchair | Washer | Oven | Laptop | Blade | Storage | Mean |
|---|---|---|---|---|---|---|---|---|---|---|---|---|
| Ang Err ↓ | Ditto [9] | 89.86 | 89.77 | 4.498 | 89.30 | 89.35 | 89.51 | 0.955 | 3.124 | 6.319 | 79.54 | 54.22 |
| | PARIS [14] | 0.069 | 0.065 | 0.019 | 0.001 | 0.020 | 0.082 | 0.028 | 0.034 | 0.001 | 0.369 | 0.069 |
| | ArtGS [18] | 0.062 | 0.034 | 0.039 | 0.038 | 0.048 | 0.081 | 0.066 | 0.052 | 0.072 | 0.020 | 0.051 |
| | REArtGS (Ours) | 0.042 | 0.059 | 0.010 | 0.006 | 0.013 | 0.067 | 0.031 | 0.012 | 0.005 | 0.201 | 0.045 |
| Pos Err ↓ | Ditto [9] | 0.201 | 5.409 | 5.698 | 1.021 | 3.768 | 0.661 | 0.129 | 0.014 | - | - | 2.113 |
| | PARIS [14] | 0.006 | 0.000 | 0.000 | 0.002 | 0.004 | 0.020 | 0.003 | 0.001 | - | - | 0.005 |
| | ArtGS [18] | 0.011 | 0.002 | 0.001 | 0.003 | 0.000 | 0.017 | 0.004 | 0.001 | - | - | 0.004 |
| | REArtGS (Ours) | 0.002 | 0.000 | 0.000 | 0.000 | 0.006 | 0.011 | 0.004 | 0.003 | - | - | 0.003 |

## D.7 Mesh Generation, Extended

To present our generation results more comprehensively, we provide the qualitative results in Fig. 10 corresponding to Table. 2 in the main draft. We generate surface meshes of the end state from the canonical state using the Gaussian deformation fields. It can be observed that our generated results still maintain smooth and high-quality surfaces, which proves the effectiveness of our method.

## D.8 Computational Efficiency Analysis for Mesh Generation

We present the detailed training time in Sec. D.2, and also provide the qualitative inference time for mesh generation in Table. 8. It can be observed that our method achieves the highest computational

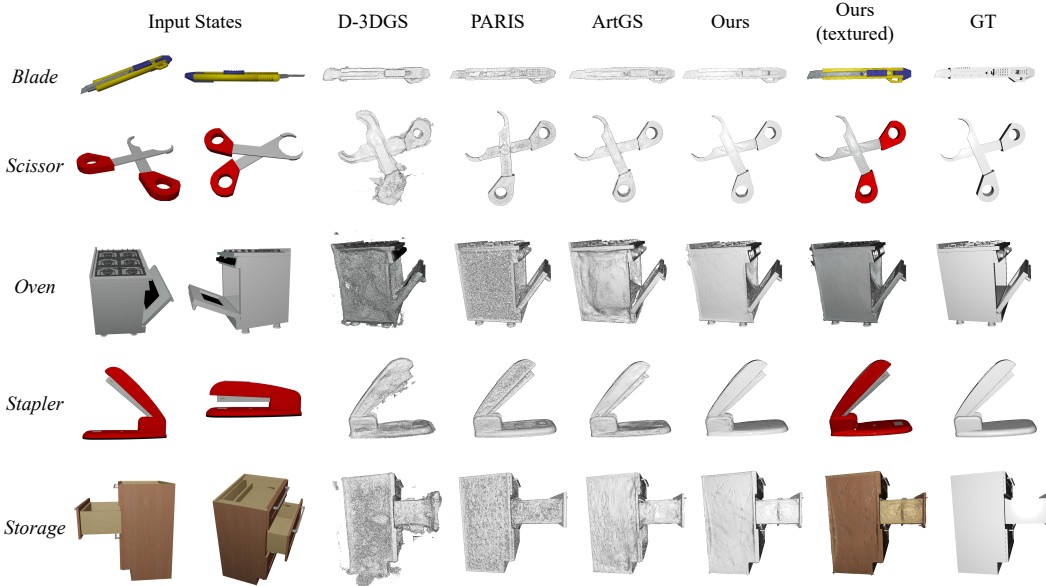

Figure 10: The qualitative results of surface generation on PartNet-Mobility dataset. We employ the deformable fields to generate surface meshes at the end state from the canonical state.

efficiency. This is mainly because of our efficient design for the learnable motion parameters, which obviates inference of deep networks.

Table 8: The qualitative results of inference time for mesh generation on PartNet-Mobility dataset. The results are derived from the average of more than ten inference times.

| Method | Time (second) ↓ |
|---|---|
| Ditto [9] | 36 |
| D-3DGS [38] | 40 |
| PARIS [14] | 33 |
| REArtGS (Ours) | **21** |

## D.9 Ablation Study, Extended

Table 9: The ablation study of canonical state setting on PartNet-Mobility dataset.

| Joint | Canonical | CD (ws) | CD (rs) | F1 | EMD |
|---|---|---|---|---|---|
| | 0.0 | **3.54** | **0.717** | **0.280** | **0.573** |
| Prismatic | 0.5 | 5.69 | 1.203 | 0.242 | 0.608 |
| | 1.0 | 6.10 | 1.304 | 0.236 | 0.591 |
| | 0.0 | 5.93 | 1.352 | 0.266 | 0.637 |
| Revolute | 0.5 | **5.88** | **1.150** | **0.275** | **0.556** |
| | 1.0 | 7.13 | 1.591 | 0.194 | 0.680 |

To evaluate the effectiveness of the canonical state selecting, we compare the surface generation quality under three settings of canonical state 0.0, 0.5, and 1.0 on PartNet-Mobility dataset. The quantitative results can be found in Table. 9. For prismatic joints, the surface reconstruction quality attains the highest when the canonical state is 0.0. It is mainly because the linear interpolation can be performed more stably taking advantage of the vector space structure of Euclidean geometry.

As for revolute joints, setting canonical state 0.5 achieves the best surface generation quality. This can be mainly attributed to the fact that canonical state 0.5 leads to the angle-bounded parameterization $\theta \in [-\pi/2, \pi/2]$, which can prevent the singularity in exponential coordinates when $\|\theta\| > \pi$.

# E   Reproducibility and Video

We provide our codes in `https://github.com/wd-ustc-cs/REArtGS` and present a video to show more visualization results in `https://sites.google.com/view/reartgs/home`.

