# OpenReview forum: "REArtGS: Reconstructing and Generating Articulated Objects via 3D Gaussian Splatting with Geometric and Motion Constraints"
_NeurIPS.cc/2025/Conference — NeurIPS 2025 poster_

### Official Review · Reviewer_Jnu3 · 2025-06-29

**Clarity:** 2
**Significance:** 2
**Originality:** 2
**Rating:** 4
**Confidence:** 4

**Summary:**

This paper purposed reartgs with can reconstruct the articulated object with given two states, and generate the internal state between the given training state. The method first extends GOF, a volume rendering based GS method, with the SDF representation to enhance the reconstruction quality. Then a deformable field is applied to the GS to predict the deformed GS for rendering.

**Questions:**

1. The author should provide more detail mentioned in the weaknesses. How to define the defomable network, and more details about generation mesh with different states.
2. More results on reconstruction datasets should be provided. Since the proposed method extends GOF with SDF, the author should also clarify why introducing SDF is important for this task, otherwise the comparison with the previous reconstruction methods is needed.

**Ethical Concerns:**

["NO or VERY MINOR ethics concerns only"]

**Final Justification:**

Although I found it is wired that the authors introduced SDF into GS ONLY to improve the reconstruction of articulated objects without comparing results with other surface reconstruction methods, the paper demonstrates SOTA performance across multiple datasets: DTU and TNT. After carefully reviewing the feedback from other reviewers and the authors' rebuttal, I am inclined to raise my overall recommendation since integrating SDF enhances reconstruction quality and benefits the community.

**Limitations:**

yes

**Quality:**

2

**Strengths And Weaknesses:**

Strengths:​​

1. Motivation is compelling. Formulating reconstruction for articulated objects (dynamic reconstruction) remains an open challenge for 3DGS.
2. The performance surpasses that of the baselines.

​​Weaknesses:​​
1. The motivation to apply an 8 layer mlp for SDF prediction is unclear. The author can provide more details about it. For example, the sdf can be a property on each GS while there is no need an implicit function to predict it as f(x) = sdf.
2. The definition of $x_{i+1}$ is missing. It might be the index of the sorted GS for each pixel.
2. Section 3.2 ("Mesh Generation with Motion Constraints") lacks clarity. Despite its title, the section does not discuss mesh generation. It describes using point clouds (based on the center of GS) to compute deformable fields, while the definition of the deformation network is missing, it might be (q,d,s)=g(x).
3. The initialization of the gs is missing. In line 189, the proposed method uses s=0.5 as the init states, and the images from s=1 used for warning up. To this end, the initialization position of the points will be important. Based on the description in line 111, the reconstruction is  built on s=0. It is confusing.
3. Since the proposed method extends GOF with SDF representation, reconstruction results on standard datasets like Tanks and Temples, DTU, and DeepDeform should be reported.

---

> ### Author Rebuttal · Authors · 2025-07-28
>
> We sincerely appreciate your valuable suggestions. The following is a point-by-point response to your questions. If you have any other questions, please let us know at the first time. We earnestly look forward to the subsequent discussions with you.
>
> **1. Motivation of Using MLP for SDF**
>
> Thank you for the insightful comments. The main motivation of using an 8-layer MLP is to map **any spatial position** $x\in \mathbb{R}^{3} $ to its corresponding SDF value. This mapping is crucial for the unbiased SDF regularization, as shown in Eq. 8. We can obtain $f(o +t^*  r)$ through the MLP and further encourage it approaches zero. Conversely, if the SDF is treated as an property of each Gaussian primitive, it is difficult to achieve point-level mapping because 3D Gaussian primitive is essentially a local probability representation. For example, if the SDF value at the center of a Gaussian primitive is taken as its SDF, the SDF value at $x = o + t^*  r$ cannot be directly obtained.
>
> **2. Explanation of $x\_{i+1}$**
>
> Thank you for your valuable comments. In fact, $x\_{i+1}$ in this paper refers to the next sampled point along the ray. We calculate  $f(x\_{i+1})$ through Taylor's first-order expansion, i,e, $f(x\_{i+1}) =f(x\_{i})+{f}'(x\_{i}) \Delta x$, where $\Delta x = \frac{z\_{\text{far}}-z\_{\text{near}}}{l} $. $f(x\_{i})$ is the SDF value of the center of the $i$-th Gaussian primitive. $z\_{\text{far}}$ and $z\_{\text{near}}$ are  the distances from the near and far cutting surfaces along the Z-negative axis to the viewpoint respectively. $l$ is set to 64. Note that we use the annealing $(1-\frac{n}{N})(\Theta (\frac{1-r\cdot f' (x\_{i})}{2} ) )-\frac{n}{N} (\Theta (-r\cdot {f}' (x\_{i}) )$ to replace ${f}'(x\_{i})$ in actual computation for adaptive transformation of view-normal angle, where $n$ and $N$ denote the current iteration and total iterations respectively. $r$ is the normalized ray direction.  $\Theta$ is the ReLU activation Function.
>
> **3. More Details of Mesh Generation and Deformable Field**
>
> Thank you for the helpful insights. We have already described the details of the deformable field in Section 3.2. Following your suggestion, we provide a more detailed elaboration on deformable field and mesh generation below. Once we finish the estimation of the joint type, we design different learnable parameters based on the joint type. For rotation joints, we define the pivot point $\mathbf{o}_{r}\in \mathbb{R}^{3}$ of the rotation axis and the normalized quaternion $\mathbf{q} \in \mathbb{R}^{4}$ as learnable parameters. For translation joints, the unit vector $\mathbf{d}\in \mathbb{R}^{3}$ of translation direction and translation distance $m$ are used as learnable parameters. Note that the deformable network in Fig. 2 is composed of these learnable parameters. We sincerely apologize for any misunderstanding we may have caused. Through Eq.10 and Eq.11, we can derive the center position $x\_{s}$ of the Gaussian primitives at any state $s$ through these learnable parameters. Therefore, the deformable field $\mathcal{g}$ can be represented as $\mathcal{g}(x)=x\_{s}$. Given any state $s$, we first obtain the deformed Gaussian primitives through the deformable field, and then render them into depth, opacity and RGB renderings. Finally we integrate these renderings into a voxel block grid and use TSDF Fusion to extract the surface mesh at state $s$. The mesh extration pipeline is described in Section 3.3.
>
> **4. Clarification on the Section 3.2**
>
> Thank you for your feedback. Although the process of mesh extraction is not directly described in Section 3.2, we elaborate on the key of mesh generation, i.e., generating accurate 3D Gaussian primitives. The dynamic texture mesh can be easily obtained by TSDF Fusion when the 3D Gaussian primitives learn the accurate motion, and the TSDF Fusion is illustrated in Section 3.3.
>
> **5. Clarification on Initialization**
>
> Thank you for your feedback. In fact, we have elaborated in detail on the initialization of Gaussian primitives of the generation state in Section 3.2. The initialization inherits the Gaussian primitives from the reconstruction stage. We define the canonical state $s^{*}=0.5$ to constrain the rotation $\theta$ within the interval $\left [ -\frac{\pi}{2}, \frac{\pi}{2} \right ] $, **preventing the singularity in exponential coordinates when $\theta > \pi$**.
>
> In the warm-up stage, we use the image supervisions from stage $s=1$ mainly for the initial part segmentation. Note that this does not affect the learning of Gaussian primitives, since the rendering loss is calculated through the deformed Gaussian primitives, which are obtained from the deformable field. Taking the rotation joint as an example, although the initialization of 3D Gaussians is used from stage $s=0$ and we define the canonical state $s^{*}=0.5$, we employ Eq. 9 and Eq. 10 to derive the deformed Gaussians and then back-propagate the rendering loss. As the rendering loss converges, the Gaussian primitives adjust to the spatial positions at state $s = 0.5$. Besides, the ablation experiments in Table. 9 also demonstrate the superiority of setting the canonical state of rotation joints to 0.5.
>
> **6. Results of More Reconstruction Datasets**
>
> Thank you for the valuable suggestions. Although our method is designed for articulated object reconstruction, it can also be generalized to common reconstruction datasets. According to your suggestion, we compared the performance on the general reconstruction datasets Tanks and Temples, DTU with the existing SOTA reconstruction methods 2DGS (Huang, 2024) and GOF. We present the following quantitative results below.
>
> **F1-score $\uparrow$ results on Tanks and Temples dataset:**
> | Scene | _Barn_ | _Caterpillar_ | _Courthouse_ | _Ignatius_ | _Meetingroom_ | _Truck_ |Mean |
> |----------------|:--------:|:---------------:|:--------------:|:------------:|:---------------:|:---------:|:---------:|
> | 2DGS       | 0.41  | 0.23         | 0.16         | 0.51       | 0.17          | 0.45    |   0.32    |
> | GOF       | 0.51   | 0.41          | 0.28         | **0.68**       | 0.28          | 0.58    |   0.46    |
> | REArtGS (Ours) | **0.63**   | **0.52**      | **0.32**     | 0.65       | **0.39**      | **0.66** |**0.53** |
>
> **CD $\downarrow$ results on DTU dataset:**
> | Scan ID | 2DGS  | GOF   | REArtGS (Ours) |
> |---------|:-------:|:-------:|:----------------:|
> | 24      | 0.48  | 0.50  | **0.37**       |
> | 37      | 0.91  | 0.82  | **0.48**       |
> | 40      | 0.39  | **0.37** | 0.45           |
> | 55      | 0.39  | 0.39  | **0.22**       |
> | 63      | 1.01  | 1.12  | **0.57**       |
> | 65      | 0.83  | 0.74  | **0.44**       |
> | 69      | 0.81  | 0.73  | **0.59**       |
> | 83      | 1.36  | 1.48  | **0.78**       |
> | 97      | 1.27  | 1.29  | **0.82**       |
> | 105     | 0.76  | 0.68  | **0.41**       |
> | 106     | 0.70  | 0.77  | **0.37**       |
> | 110     | 1.40  | 0.90  | **0.61**       |
> | 114     | 0.40  | 0.42  | **0.33**       |
> | 118     | 0.76  | 0.66  | **0.31**       |
> | 122     | 0.52  | 0.49  | **0.38**       |
> | Mean    | 0.80  | 0.74  | **0.48**       |
>
> **7. The Significance of Introducing SDF**
>
> Opacity constitutes a fundamental attribute for the $\alpha$-blending in 3DGS. However, since the opacity cannot explicitly define the scene surface, opacity-based 3DGS methods typically yield surface mesh results with artifacts. For example, GOF extracts surface mesh through an opacity level set, but the Gaussian opacity field poses non-strict linearity, leading to mesh extraction with holes. In contrast, SDF has strong linearity and explicitly defines the scene surface. We hence aim to introduce the SDF to regularize the Gaussian opacity field. Concretely, we propose the unbiased regularization, which encourages Gaussians near the surface to exhibit a high opacity value, eliminating artifacts distant from the surface.
>
> On the other hand, due to the irregular nature of 3D Gaussians, it is difficult to estimate their normal directions. Existing methods, such as 2DGS and PGSR, attempt to facilitate the normal estimation by compressing 3D Gaussian primitives to 2D shapes. This geometric constraint is not reasonable as the restriction on the shape of Gaussian primitives leads to degraded performance in capturing detailed structures. In contrast, our method introduces SDF in 3D Gaussian primitives and leverages the gradients of SDF to accurately define the normals, instead of restricting the shapes of Gaussian primitives.

---

> > ### Author Response · Authors · 2025-08-02
> >
> > Dear reviewer, thank you for your thoughtful and constructive opinions. We have carefully responsed all your comments and provided point-by-point responses. If you have any further questions or need additional clarification, please let us know before the discussion ends. We truly appreciate your time and support.

---

> > ### Comment · Reviewer_Jnu3 · 2025-08-02
> >
> > Thanks for the response.
> >
> > Could you please report more details about the exp on the tnt and dtu? Like the running time and the number of iterations.

---

> > > ### Author Response · Authors · 2025-08-05
> > >
> > > Dear reviewer, thank you so much for your thoughtful feedback.  We sincerely hope that our response has addressed your concerns. As the discussion deadline approaches in just three days, if you still have any questions or need further clarification, we would be truly delighted to provide additional theoretical analysis or experimental results. Moreover, if your concerns have already been resolved, we would greatly appreciate it if you could kindly share your feedback with us. Your recognition means a great deal to us, and we will be deeply grateful for receiving your recognition.

---

> ### Author Response · Authors · 2025-08-02
> **Detailed implementation of TNT and DTU dataset**
>
> Thank you for your timely reply. Our method is trained for 30,000 iterations on both Tanks and Temples and DTU datasets. Specifically, we only conduct the reconstrucion stage since neither Tanks and Temples dataset nor DTU dataset imposes any generation requirements for articulated objects. We omit the additional 3,000 warm-up iterations for SDF learning as we observe that they yield slight improvement on actual performance. We perform the densification and pruning of 3D Gaussians from 500 to 15,000 iterations and apply SDF-normal regularization and Eikonal regularization from 500 to 30,000 iterations, the unbiased SDF regularization is employed from 500 to 15,000 iterations.
>
> The average training time on the Tanks and Temples and DTU dataset is 40 minutes and 30 minutes respectively. Note that we have conduct engineering optimizations to our method, which can be summarized as following. (1) We calculate $f(x\_{i+1})$ through Taylor's first-order expansion, i,e, $f(x\_{i+1}) =f(x\_{i})+{f}'(x\_{i}) \Delta x$. In previous experiments, we define $\Delta x = RS\cdot r$, where $R\in \mathbb{R}^{3\times3}$ is the R is the rotation matrix derived from Gaussian’s quaternion, and $S\in \mathbb{R}^{3}$ represents Gaussian's scaling. $r$ is the normalized ray direction. However, we observe that this calculation consumes a lot of time and it is difficult to obtain $\Delta x$ when the object size is extremely small. Therefore, we have refined the calculation to $\Delta x = \frac{z\_{\text{far}}-z\_{\text{near}}}{l} $,  where $z\_{\text{near}}$, $z\_{\text{far}}$ are  the distances from the near and far cutting surfaces along the Z-negative axis to the viewpoint respectively. $l$ is set to 64. This approach not only remarkably enhances the computational efficiency, but also demonstrates better robustness against extremely small objects. The average cd(ws) $\downarrow$ reduces -0.008 compared with previous results. (2) Inspired by PARIS, we also employ the Tiny-cuda-nn to accelerate our SDF network, achieving a significant time reduction. Besides, this acceleration have almost no impact on the mesh results.
>
> Moreover, our approach still maintains SOTA results with the acceleration on the PartNet-Mobility dataset. Please refer to the response to reviewer BVdL for details.

---

> > ### Author Response · Authors · 2025-08-02
> > **Supplementary instruction of implementation on TNT and DTU dataset**
> >
> > To ensure the fairness of the experiment, GOF, 2DGS and our method are all trained for 30,000 iterations on the Tanks and Temples and DTU datasets. Moreover, our training time is close to GOF. On a single RTX 4090 GPU, our training time is about 40 minutes and 30 minutes on Tanks and Temples and DTU datasets respectively. In contrast, GOF requires 36 minutes and 28 minutes on Tanks and Temples and DTU datasets approximately.

---

### Official Review · Reviewer_BVdL · 2025-07-02

**Clarity:** 3
**Significance:** 3
**Originality:** 3
**Rating:** 5
**Confidence:** 4

**Summary:**

This paper tackles an interesting task: generation articulatable objects from multiview observations collected in two distinct states of the object articulation.
It improves upon prior work PARIS and ArtGS. The former uses a NeRF representation, and the latter uses Gsplats but requires RGB-D images whereas the proposed methos REArtGS needs only RGB images.
The method differs from prior work by providing a much more accurate mesh reconstruction in the initial state by adopting ray-traced GS formulation (Gaussian Opacity Field) and connecting the that opacity with a jointly learned SDF.
Secondly, during the articulation learning step, it also incorporated heuristic, but effective, rules to correctly determine the moving parts and their articulation type.
The paper showed impressive improvements compared to prior work in both synthetic Part-Net data and real-world captured data.

**Questions:**

I'm curious what is the fundamental difference between a NeRF-based approach like PARIS and 3DGS-based approach, such as REArtGS?
The paper focused on introducing geometry regularization specifically for 3DGS. But what benefit do we get by using 3DGS? If I used NeuS instead of NeRF in PARIS (thus no need these additional techniques introduced in this paper) and if I applied similar deformation optimization, would I get similar result?
Is there any gains in terms of training speed, texture quality etc?

Secondly, I couldn't find how many images are required for optimizations in the two stages.

Lastly, how does the method compare with other work in terms of training time?

**Ethical Concerns:**

["NO or VERY MINOR ethics concerns only"]

**Final Justification:**

The rebuttal sufficiently addressed my question. I remain positive about the paper.

**Limitations:**

I think a key limitation, not restricted to this paper alone, is potential failure when the observations of the initial state is incomplete.
In this case, one may think of combining the second observed state and jointly optimize the reconstruction and deformation.
It might be worthy mentioning this in the paper.

**Paper Formatting Concerns:**

None.

**Quality:**

3

**Strengths And Weaknesses:**

# Strength
- The paper is clearly written. Moreover, the authors ensure the reproducibility of their implementation by stating that core modules and generation inference codes are provided in the supplementary materials.
- The paper presents convincing visual and quantitative improvements upon prior art. Qualitatively, REArtGS demonstrates "significantly smoother and clearer surfaces" for reconstruction

# Weaknesses
- I find the reporting of the tables and figures curiously unmatched. Why do the figures report D-3DGS while the tables don't even include this method? The close contender ArtGS is not shown only one figure in the Appendix.
- Since the paper mentioned multi-part and outlined a solution in the appendix, I would like to see a couple of examples. Otherwise, the paper feels rather incomplete.
- Motivation of using 3DGS (see my questions below).
- The training time seems rather long - 95 minutes in total. There is no reference to the training time of prior work.

# Minor weaknesses
- For texture extraction, I'm curious if the texture considers only SH0 and discard the rest of the SH degrees? If so, it might be better to mention if the reconstruction is done with only sh0.

---

> ### Author Rebuttal · Authors · 2025-07-29
>
> We sincerely appreciate your recognition of our method. The following is a point-by-point response to your questions. If you have any other questions, please let us know at the first time. We sincerely look forward to the subsequent discussion with you.
>
> **1. Motivation of Using 3DGS**
>
> Thanks for the insightful opinions. NeRF and NeuS represent 3D scenes through neural implicit radiance fields, posing four major limitations. (1) The implicit representation tends to yield entangled neural representations, leading to the shape-radiance ambiguity.
> (2) Their ray marching sampling may skip over thin structures, resulting in unsatisfactory reconstruction of details. (3) When extracting surface meshes from NeRF-based methods, post-processing algorithms like Marching Cubes are typically required. However, using Marching Cubes from the neural representations often introduces unconsidered errors, producing noisy surface mesh results. (4) The texture is strongly coupled with the implicit representation, and high-frequency textures rely on positional encoding, which is prone to causing over smoothing.
>
> In contrast, our motivation for adopting 3DGS is based on the following benefits. (1) 3DGS is an explicit point-based representation, which is inherently more consistent to surface mesh reconstruction. Through the anisotropic Gaussian primitives, 3DGS can precisely recover detailed structures. (2) The Spherical Harmonics (SH) of 3D Gaussian primitives support direct texture baking for texture extraction. For a detailed discussion on SH, please refer to Point 5. (3) In terms of training time, 3DGS uses the rasterization instead of volume rendering, significantly improving the training speed. For example, NeuS, which also introduces SDF, requires more than 12 hours. Please refer to Point 3 for details. (4) Leveraging the explicit nature of 3DGS, we can explicitly obtain the deformed Gaussian primitives through deformable fields. In contrast, Nerf-based methods like PAIRS require composite neural radiance fields for the two-stage inputs. Since spatial overlap exists between the composite density fields of the start and end states, the volume rendering with neural implicit representation produces noisy surface reconstruction results. As shown in Fig. 5, PARIS yields unsmooth surface mesh results while our method achieves high-quality detailed reconstruction. Therefore, **the degraded mesh results stems from the limitations of the neural implicit radiance field and cannot be resolved merely by using NeuS.**
>
> **2. Input View Setting**
>
> We follow the same input settings as PARIS and ArtGS, using 64 multi-view images and 50 test images per stage. We have also supplemented an ablation study on the number of input images from each stage below. We report the average surface mesh generation results on PartNet-Mobility dataset.
> | Views | CD (ws)$\downarrow$ | CD (rs)$\downarrow$  | F1$\uparrow$ | EMD$\downarrow$ |
> |--------|-------|-------|-------|-------|
> | 64 | **4.10** | **1.063** | **0.298** |**0.670** |
> | 32 | 4.35 |1.284  | 0.280  |  0.743 |
> | 16 | 6.73 | 1.510 | 0.201 |  0.857 |
> | 8 | 48.21 | 4.652 | 0.074 |  1.552 |
>
> The experimental results demonstrate that even if the number of input images per stage is decreased to 16, our method still exhibits robust surface generation results.
>
> **3. Comparison of Training Time**
>
> We sincerely appreciate your invaluable comments. In the early experiments, the training process takes 65 minutes for reconstruction and 30 minutes for generation. Nevertheless, we have now implemented engineering optimizations in our approach, which significantly reduce the training time to approximately 30 minutes for reconstruction and approximately 10 minutes for generation. Moreover, Our approach still maintains SOTA results with the acceleration. Specifically, our main engineering optimizations can be summarized as following. (1) We calculate $f(x\_{i+1})$ through Taylor's first-order expansion, i,e, $f(x\_{i+1}) =f(x\_{i})+{f}'(x\_{i}) \Delta x$. In previous experiments, we define $\Delta x = RS\cdot r$, where $R\in \mathbb{R}^{3\times3}$ is the rotation matrix derived from Gaussian’s quaternion, and $S\in \mathbb{R}^{3}$ represents Gaussian's scaling. $r$ is the normalized ray direction. However, we observe that this calculation consumes a lot of time and it is difficult to obtain $\Delta x$ when the object size is extremely small. Therefore, we have refined the calculation to $\Delta x = \frac{z\_{\text{far}}-z\_{\text{near}}}{l} $, where  $z\_{\text{far}}$ and $z\_{\text{near}}$ are the distances from the near and far cutting surfaces along the Z-negative axis to the viewpoint respectively. $l$ is set to 64. This approach not only remarkably enhances the computational efficiency, but also demonstrates better robustness against extremely small objects. For PartNet-Mobility data in the paper, the average cd(ws) $\downarrow$ reduces -0.008 compared with previous results. (2) Inspired by PARIS, we also employ the Tiny-cuda-nn to accelerate our SDF network, achieving a significant time reduction. Besides, this acceleration have almost no impact on the mesh results. (3) We observe that our method can stop early at 20,000 iterations during the generation stage, and the mesh quality only suffers a slight degradation (approximately +0.1 at average cd(ws) $\downarrow$).
>
> Following your suggestion, we present a quantitative comparison for the average training time and the average CD(ws) results on PartNet-Mobility dataset, shown as following. “acc” denotes the aforementioned acceleration, and “Recon.” and “Gen.” are reconstruction and generation respectively.
>
> | Methods | Time $\downarrow$ | Recon. CD (ws)$\downarrow$  | Gen. CD (ws)$\downarrow$  |
> |--------|-------|:-:|:-:|
> | NeuS | >12h | 4.43 | -|
> | GOF | **28 min** | 6.09 | - |
> | PARIS | 45 min |3.94  | 0.280  |  8.16 |
> | ArtGS | 30 min | 4.75 | 0.201 |  4.92 |
> | Ours w/ acc | 40 min | **3.71** | 4.20 |
> | Ours w/o acc | 95 min | 3.79 | **4.10** |
>
> **4. Two-stage Joint Optimization**
>
> We sincerely appreciate your professional review comments. Although we adopt a two-stage optimization strategy, the data of each stage is not isolated. In fact, the two-stage image supervision can mutually compensate for each other's unseen regions. Since initialization of the generation stage inherits the Gaussian primitives from the reconstruction stage, the incomplete observation regions in the reconstruction stage can be effectively optimized by the image supervisions of the generation stage. Note that since the deformable field acts solely on movable parts, the rendering loss during the generation stage can be mapped to reasonable pixel coordinates.
>
> Moreover, if the viewpoints in the generation stage do not completely cover the object, the reconstruction stage enables to provide a high-quality initialization for generation. To further validate the analysis, we have conducted ablation experiments with different input views for the two stages, shown as following. “View-1” and “View-2” denote the views of reconstruction and generation stage respectively.
>
> | Views-1 | Views-2 | Recon. CD (ws)$\downarrow$ | Gen. CD (ws)$\downarrow$  |
> |--------|-------|-------|-------|
> | 64 | 64 | **3.79** | **4.10** |
> | 64 |16| **3.79** | 5.46 |
> | 16 | 64|6.48|5.10 |
> | 16 |16|  6.48 | 6.73 |
>
> The experimental results demonstrate that when one stage utilizes incomplete observation data (16 random sampled views) and the other stage offers more supervisions (64 views), the mesh results exhibit improvement compared to the case where both stages employ incomplete observations, demonstrating our analysis.
>
>
> **5. The SH coefficient in texture extraction**
>
> Thank you for your professional technical review. This is an interesting question. In fact, we first use all the Spherical Harmonics (SH) to obtain view-dependent rendered images when extracting textured meshes. However, The TSDF fusion only bakes the static RGB renderings, which is equivalent to SH0. This design aligns with two core objectives: (1) Higher-order SH coefficients with view-dependent effects are challenging to bake into fixed textures. (2) The view-dependent effects encoded by higher-order SH terms are spatially variant and non-transferable to novel lighting conditions. Retaining them in the static texture maps would introduce artifacts when relighting the mesh.
>
> **6. Questions on Table and Figure**
>
> Since Deformable-3DGS (D-3DGS) is a dynamic generation method, we only compare the mesh generation results with it. The qualitative results are shown in Fig 5 and 10, corresponding to the quantitative data in Table 2. ArtGS is essentially an RGBD image-based method. In the main draft, we have already compared its surface mesh reconstruction and generation results without depth inputs with our method to demonstrate our superiority on the core topic of this paper. In the supplementary material, we aim to further demonstrate the completeness of our method REArtGS, including part segmentation and joint estimation. Thus we mainly present the visualization results of our method in the supplementary material. Following your suggestions, we have finished additional qualitative comparisons with ArtGS in part segmentation and joint estimation and will supplement them after acceptance, due to the limitations on images and external links in this rebuttal.  Meanwhile, please refer to the response to Reviewer E8MT for more quantitative comparison results with ArtGS, due to character limitations.
>
>
> **7. Qualitative Results of Multi-Part Articulated Objects**
>
> Thank you for your feedback. We have already presented the high-quality reconstruction and generation results of an articulated object with three movable parts at the fourth row of Figure 6 in the main draft. More qualitative results of multi-part articulated objects will be presented after acceptance, due to the limitation of uploading images and external links.

---

> > ### Comment · Reviewer_BVdL · 2025-08-05
> > **Is the first table wrong?**
> >
> > Thank you for your response and new experiments. I appreciate the detailed explanation motivating the use of GSplats. The first table in the responses show very good CD for PARIS and ArtGS in the last column. How should I interpret that?

---

> > > ### Author Response · Authors · 2025-08-05
> > >
> > > We sincerely thank you for pointing out our mistakes. We apologize for our carelessness. The CD (ws) results of mesh generation of ArtGS and PARIS were recorded as incorrect values due to our negligence. **The correct results for the average CD (ws) on mesh generation of ArtGS and PARIS are already shown in Table 2 in the main draft, which are 4.92 and 8.16 respectively.** Now we have updated the table to the correct results.
> > >
> > > We sincerely thank you for your professional review and the time you have devoted. If you have any other concerns, please let us know immediately. We would be truly delighted to provide additional theoretical analysis and experimental results to clarify any points of your concern. Your feedback is very important for our work REArtGS.

---

> > > > ### Author Response · Authors · 2025-08-05
> > > > **Correction of Additional Experiments**
> > > >
> > > > We provide the corrected experimental results for additional comparison of training time in the following table. “Recon.” and “Gen.” are reconstruction and generation respectively. We report the average results on PartNet-Mobility dataset.
> > > >
> > > > | Methods | Time $\downarrow$ | Recon. CD (ws)$\downarrow$  | Gen. CD (ws)$\downarrow$  |
> > > > |--------|-------|-------|-------|
> > > > | NeuS | >12h | 4.43 | - |
> > > > | GOF | **28 min** | 6.09 | - |
> > > > | PARIS | 45 min |3.94   |  8.16 |
> > > > | ArtGS | 30 min | 4.75  |  4.92 |
> > > > | Ours w/ acc | 40 min | **3.71** | 4.20 |
> > > > | Ours w/o acc | 95 min | 3.79 | **4.10** |

---

> > ### Comment · Reviewer_BVdL · 2025-08-05
> > **accept**
> >
> > The rebuttal addressed my questions sufficiently. The acceleration from the authors seems to be an important advantage over existing work, without, I would be hesitant to accept it.

---

### Official Review · Reviewer_oBKh · 2025-07-02

**Clarity:** 3
**Significance:** 2
**Originality:** 2
**Rating:** 4
**Confidence:** 4

**Summary:**

This paper aims to solve the problems of (1) 3D textured surface reconstruction of articulated objects from multiview RGB images (2) generating dynamic unseen motion states from a given start state to an end state both of which have multiview RGB images. The paper proposes a novel approach using unbiased Signed Distance Field (SDF) guidance for regularization of Gaussian opacity fields for solving the problem in (1). Reconstruction and Motion Generation results are demonstrated on a synthetic dataset and a real world dataset.

**Questions:**

1. What is $x_{i+1}$ in Eq. 2 ? In NeuS i refes to the index of the sampled point along the ray, in this case the how is the (i+1)th gaussian defined with respect to the ith Gaussian?

2. How many input views are used for each of the two states (both source and target) ?

3. Lines 94-95 : Although improving the surface reconstruction quality, these methods still lack “more reasonable geometry constraints”. Exactly what is unrealistic or unreasonable about the geometric constraints in GOF, 2DGS and PGSR that this paper has improved?

**Ethical Concerns:**

["NO or VERY MINOR ethics concerns only"]

**Final Justification:**

Thanking the authors for their detailed justification. Based on the authors' feedback, I have updated my final rating to 4.

**Limitations:**

Yes

**Quality:**

3

**Strengths And Weaknesses:**

Strengths:
+ The paper attempts to solve a challenging intermediate state generation problem given multiview images of a only start and end state of an articulated object, and achieves geometry and texture generation of reasonably good quality on real world datasets and on multi-part articulated objects.
+ The paper is well-written and easy to follow.
+ The geometry reconstruction results on USB and Fridge in Fig.4 and the generation results in Fig. 5 clearly denote better geometric quality.

Weaknesses :

1. Originality : This paper reuses concepts and ideas from NeuS (Eq. 2), GOF (Eq. 3-5) and 3DGSR (bell-shaped function similar to Eq. 6).  SDF guidance for 3D Gaussian primitives has been done in 3DGSR.

2. Results :
- Fig 4 – The reconstruction quality of Washer and Laptop does not seem to be very good.
- Missing Qualitative comparison with state-of-the-art methods on real-world dataset AKB-48.
- Qualitative comparison with Ditto missing in Fig 4.
- Qualitative ablation results needed to understand the significance of the proposed geometric constraints.
- Quantitative / qualitative comparison of 3D reconstruction not done with 3DGSR, which also uses SDF along with 3DGS for unified surface and appearance modeling, and uses a similar bell-shaped function.


3. Writing Ambiguity : Lines 94-95 : Although improving the surface reconstruction quality, these methods still lack “more reasonable geometry constraints”.



4. Undefined notations:

- $x_{i+1}$ in Eq. 2 not defined.

 - $x_L$ and $t$ in Eq. 3 are not defined.

---

> ### Author Rebuttal · Authors · 2025-07-27
>
> We sincerely appreciate your valuable comments. The following is a point-by-point response to your questions. If you have any other questions, please let us know at the first time. We sincerely look forward to the subsequent discussion with you.
>
> **1. Question on $x\_{i+1}$ and Undefined Symbols**
>
> Thank you for your insightful opinions. In fact, $x\_{i+1}$ in this paper refers to the next sampled point along the ray. We calculate  $f(x\_{i+1})$ through Taylor's first-order expansion, i,e, $f(x\_{i+1}) =f(x\_{i})+{f}'(x\_{i}) \Delta x$, where $\Delta x = \frac{z\_{\text{far}}-z\_{\text{near}}}{l} $. $z\_{\text{far}}$ and $z\_{\text{near}}$ are  the distances from the near and far cutting surfaces along the Z-negative axis to the viewpoint respectively. $l$ is set to 64. Note that we use the annealing $(1-\frac{n}{N})(\Theta (\frac{1-r\cdot f' (x\_{i})}{2} ) )-\frac{n}{N} (\Theta (-r\cdot {f}' (x\_{i}) )$ to replace ${f}'(x\_{i})$ in actual computation for adaptive transformation of view-normal angle, where $n$ and $N$ denote the current iteration and total iterations respectively. $r$ is the normalized ray direction.  $\Theta$ is the ReLU activation Function.
> We also apologize for the undefined symbols. $x\_{L}$ and $t$ represent the spatial position of Gaussians local coordinate system and ray depth respectively.
>
> **2. Input View Setting**
>
> We follow the same input settings as PARIS and ArtGS, using 64 multi-view images and 50 test images per stage. We have also supplemented an ablation study on the number of input images.  Due to character limitations, please refer to the rebuttal provided to Reviewer BVdL for details.
>
> **3. Question on the Geometry Constraints of GOF, 2DGS and PGSR**
>
> Thank you for your feedback. 2DGS, PGSR, and GOF are all essentially based on the Gaussian opacity fields. Among them, 2DGS and PGSR attempt to facilitate the normal estimation by compressing 3D Gaussian primitives to 2D shapes. This geometric constraint is not reasonable as the restriction on the shape of Gaussian primitives leads to degraded performance in capturing detailed structures. On the other hand, GOF attempts to optimize the rendering process of 3DGS and approximates the Gaussian’s normal as the normal of the ray-Gaussian intersection plane. The approximation typically introduces normal estimation errors. Moreover, since Gaussian opacity field cannot explicitly define the scene surface, the opacity-based methods often extracts surface meshes with artifacts distant from the surface. Therefore, they still lack more reasonable geometric constraints.
>
> In contrast, our method introduces SDF in 3D Gaussian primitives and leverages the gradients of SDF to accurately define the normals, instead of restricting the shapes of Gaussian primitives or using approximation strategy. And the unbiased regularization encourages Gaussians near the surface to exhibit a high opacity value, achieving more reasonable geometric constraints.
>
> **4. Clarification on Originality**
>
> We sincerely appreciate your dedicated efforts in the research of the relevant works. We first clarify that this paper does not merely reuse the concepts from Neus, GOF, and 3DGSR. Instead, our REArtGS is the first framework that innovatively introduces SDF to regularize the Gaussian opacity field, which plays a core attribute in 3DGS rendering pipelne.
>
> Among the existing methods, GOF is an opacity-based 3DGS method. The Eq. 3 and Eq. 5 in this paper fundamentally differ from GOF. Specifically, GOF solely determines the $\alpha$-blending weight by the rendering contribution $\varepsilon$, which fails to explicitly define the relationship between Gaussian primitives and the scene surface, leading to results with artifacts. In contrast, our method computes the $\alpha$-blending weight based on both  SDF $\sigma(f(x))$ and the rendering contribution $\varepsilon$ (see Eq. 2, Eq. 3, and Eq. 5). The points with the maximum rendering contribution are aligned with the scene surface through the unbiased SDF regularization, enhancing surface reconstruction quality. Note that we quote Eq.4 for the detailed elaboration of the unbiased SDF regularization.
>
> Although 3DGSR also incorporates SDF with 3DGS, its $\alpha$-blending weights purely depend on the SDF-derived opacity, which only utilizes Gaussians’ center as input and ignores their local features.  In contrast, our REArtGS integrates both local and global information through Eq. 2 and Eq. 3. Conretely, Eq. 3 combines the rendering contribution derived from local properties of Gaussian primitives with the SDF encoding global spatial position information.
>
> Besides, Neus is built upon a neural implicit continuous representation, allowing the continuous SDF to be seamlessly incorporated into its rendering pipeline. In contrast, 3DGS is a discrete point-based rendering method, making direct adoption of SDF infeasible. This constitutes the fundamental distinction between our method and Neus.
>
> In summary, Our REArtGS has essential differences from these methods at the theoretical level and has achieved substantial improvements rather than simple reuse or combination.
>
> **5. Flawed Qualitative Results**
>
> Thank you for your insightful comments on our paper. In fact, the reconstruction of self-luminous screens and flat surfaces lacking texture features is a non-trivial task for current surface reconstruction technique. These two cases impose strict requirements on the geometric flatness of the reconstruction results and pose significant challenge for existing methods. In future work, we will introduce illumination decomposition to address self-luminous materials and enhance the planar constraints of 3DGS through RANSAC to enhance the reconstruction of flat surfaces lacking features
>
> However, our method still achieves the best CD(rs) and EMD scores on washer and laptop,  and accurately recovers the global structure as shown in Fig. 4. We also observe that key parts such as the keyboard and the washing machine lid are all reconstructed with high quality. Therefore, these local imperfections in the reconstructions do not affect the actual manipulation of articulated objects.
>
> **6. Missing Qualitative Comparison**
>
> We sincerely appreciate your invaluable feedback. Following your suggestions, we have already finished the qualitative comparisons with other methods on AKB-48 datasets, the qualitative comparison with Ditto, as well as the qualitative ablation results of geometric constraints.
>
> Unfortunately, given the limitation that images and external links cannot be uploaded during this rebuttal, we sincerely apologize for not being able to share these qualitative results with you at the first time, and we formally promise to add these results in this paper after accepted.
>
> Moreover, the content already presented in this paper is sufficient to illustrate the aforementioned issues. Firstly, we conduct extensive comparisons on synthetic datasets, comprehensive demonstrating the superiority of our method. For real-world experiments, our qualitative results aim to fully showcase the generalization ability of the method on real-world data, and the quantitative results in Table. 5 further confirm our performance advantages. Even though the qualitative results of Ditto are not presented, we present the qualitative comparisons with A-SDF, which also uses 3D supervision. And we have conductd a quantitative comparison with Ditto in Table. 1, which already indicates our superiority.  Finally, we have presented the qualitative ablation results of unbiased SDF regularization at distribution level in Fig. 3, and the quantitative ablation results of geometric constraints in Table. 3, validating the effectiveness of proposed geometric constraints.
>
> **7. Comparison with 3DGSR**
>
> We appreciate the time you devoted to the related works. Although 3DGSR also uses SDF and a similar bell-shaped function as ours, we are sorry that we cannot provide a comparison with it because its codes are not available and its official experimental results on articulated objects are not provided. Based on your valuable suggestions, to further demonstrate the effectiveness of our unbiased SDF regularization, we have conducted a quantitative comparison with GSDF (Yu, 2024), which also introduces SDF in 3DGS, shown as following:
> | Metrics   | Method          | Stapler  | USB     | Scissor  | Fridge   | Foldchair | Washer   | Blade    | Laptop   | Oven     | Storage  | Mean     |
> |-----------|-----------------|----------|---------|----------|----------|-----------|----------|----------|----------|----------|----------|----------|
> | CD(ws)↓   | GSDF            | **2.96** | 10.23   | 0.64     | 2.99     | 0.81      | 19.54    | 0.85     | 0.90     | 22.17    | 9.38     | 7.05     |
> |           | REArtGS(Ours) | 3.47 | **0.75** | **0.29** | **1.50** | **0.40**  | **12.20** | **0.72** | **0.53** | **8.89** | **8.27** | **3.79** |
> | CD(rs)↓   | GSDF            | 2.732    | 3.810   | 2.856    | 2.506    | 1.146     | 2.142    | 3.571    | 0.708    | 3.053    | 2.704    | 2.523    |
> |           | REArtGS(Ours) | **2.186** | **1.433** | **2.291** | **0.475** | **0.180**  | **1.204** | **2.596** | **0.038** | **0.784** | **1.330** | **1.236** |
> | F1↑       | GSDF            | 0.232    | 0.146   | 0.435    | 0.075    | 0.305     | 0.042    | 0.358    | 0.367    | 0.024    | 0.046    | 0.203    |
> |           | REArtGS(Ours) | **0.256** | **0.307** | **0.598** | **0.165** | **0.502**  | **0.069** | **0.488** | **0.419** | **0.065** | **0.066** | **0.294** |
> | EMD↓      | GSDF            | 1.715    | 1.760   | 1.105    | 1.067    | 0.530     | 1.334    | 1.630    | 0.210    | 0.642    | **0.719**  | 1.071    |
> |           | REArtGS(Ours) | **1.060** | **0.847** | **1.078** | **0.485** | **0.097**  | **0.777** | **1.112** | **0.111** | **0.627** | 0.755 | **0.695** |

---

> > ### Author Response · Authors · 2025-08-02
> >
> > Dear reviewer, thank you so much for your thoughtful feedback. We have carefully gone through each of your comments and provided detailed responses. If there is anything we may have missed or if you would like us to clarify anything further, please don’t hesitate to let us know before the discussion concludes. Your insights are truly valuable to us, and we deeply appreciate your support.

---

> > ### Author Response · Authors · 2025-08-05
> >
> > Dear Reviewer, thank you sincerely for the time and thoughtful effort in reviewing. As the discussion deadline approaches in just under three days, we would like to kindly ask whether you have any additional questions or if there are any further experiments you would like us to consider. We are truly happy to provide additional theoretical analysis or experimental results as soon as possible to help clarify any points of concern. Your feedback is truly valuable to our work REArtGS, and we would be grateful if we could receive your recognition.

---

### Official Review · Reviewer_zVCD · 2025-07-03

**Clarity:** 3
**Significance:** 3
**Originality:** 3
**Rating:** 5
**Confidence:** 4

**Summary:**

The authors propose a solution to surface reconstruction and generation for articulated objects by coupling geometric and motion constraints to 3D Gaussian primitives.
Given multi-view RGB images of arbitrary two states of articulated objects, the method utilizes an unbiased Signed Distance Field (SDF) guidance to regularize Gaussian opacity fields.
It then establishes deformable fields for 3D Gaussians, constrained by the kinematic structures of articulated objects, thereby achieving unsupervised generation of surface meshes in unseen states.

**Questions:**

What is the effect of the diversity of the two views on surface reconstruction quality? Does the method require the two surfaces to be very distinct, and would the surface reconstruction suffer if the images are close?

Can the method take more than two input images? Let's say $s_0=0$, $s_1=0.1$ and $s_2=1.0$, would that help the reconstruction quality or would it have any effect?

**Ethical Concerns:**

["NO or VERY MINOR ethics concerns only"]

**Limitations:**

- The method requires accurate camera poses.
- Rendering of objects with transparent materials and non-Lambertian surfaces are challenging.

**Paper Formatting Concerns:**

Formatting looks good.

**Quality:**

3

**Strengths And Weaknesses:**

**Strength**
- The proposed method utilizes unbiased SDF guidance for 3D Gaussian primitives to enhance geometric constraints, thereby improving the reconstruction quality.
- It learns deformable fields and generates unseen states by constraining the kinematic structures of articulated objects.
- The method is evaluated on different synthetic and real datasets.

**Weaknesses**
- The method requires access to posed images as input.

---

> ### Author Rebuttal · Authors · 2025-07-29
>
> We sincerely thank you for your recognition and encouragement of our method. The following is a point-to-point response to the questions you raised. If you have any other questions, please let us know at the first time. We sincerely look forward to the subsequent discussion with you.
>
> **1. The Effect of The Diversity of The Two-Stage Views**
>
> We sincerely appreciate your professional review comments. Our approach exhibits remarkable robustness to the diversity of two-state views. Specifically, our method enable realistic surface mesh generation unless an articulated object undergoes extremely minimal rotation or translation.
>
> To validate the robustness of our method on the diversity of two-state views, we have conducted ablation experiments on the rotation angles and translation distances in the second state for USB and storage of the PartNet-Mobility dataset.  Concretely, we adjust the rotation angle and translation distance of articulated object at the second state to different values, and then generates the surface mesh at $s=0$ from the canonical state. We report  the CD (ws) results below.
>
> |         | Storage |      |    |           | USB |            |
> |:-:|:-:|:-:|:-:|:-:|:-:|:-:|
> | 0.3m | 0.6m | 0.9m | |15°| 30°| 45°|
> |**7.43** | 7.71 | 7.48 ||1.43 | 1.52 | **1.35**|
>
> We observe that the diversity of two-state viewpoints does not lead to a significant influence on the mesh generation results of our method. For the storage, even a smaller translation distance yields superior surface mesh generation. This demonstrates the robustness of our method against the diversity of two-state views.
>
> **2. Question on Taking More State Inputs**
>
> Thank you for your valuable comment. This is an interesting question. During the generation stage, our method use the deformed Gaussians obtained from the deformable field to calculate the rendering loss with the image supervisions at corresponding state. Therefore, incorporating view inputs from additional states is equivalent to adding more image supervisions,  which can naturally enhance the surface mesh generation quality.
>
> We also validate this analysis on the USB of PartNet-Mobility dataset. Specifically, we introduce additional view inputs at the state $s = 0.1$. As a result, the CD (ws) result of surface mesh generation improves significantly from 1.68 to 0.85. This proves that our method can improve the mesh generation quality through additional state view inputs.
>
> Moreover, we have demonstrated that views from different states mutually compensate for each other's unseen regions. For more details, kindly refer to our response to Reviewer BVdL.
>
> **3. Reliance on Camera Poses**
>
> Thank you for your valuable feedback. In fact, camera pose estimation is now a ready-to-use technology. For multi-view images, our method can directly use Colmap to estimate the corresponding camera poses. The camera poses estimated by Colmap are sufficient for our method to achieve high-quality surface reconstruction and generation. In recent years, many excellent camera pose estimation methods have also been able to meet our requirements for camera poses, such as Reloc3r (Dong, 2024). Therefore, we believe that the reliance on camera poses does not constitute an intractable shortcoming, and we aim to introduce Reloc3r into our framework for camera pose estimation in the future work.
>
> **4. Rendering of Objects with Transparent Materials and Non-Lambertian Surfaces**
>
> Thank you for the insightful comments. Transparent materials and non-Lambertian surfaces undoubtedly pose challenges in current 3D reconstruction due to their complex requirements for light modeling. However, several works attempt to address the reconstruction of transparent materials and non-Lambertian surfaces recently, and we are inspired by them. In the future works, we aim to introduce the Gaussian light field probes to encode both ambient light and specular refraction for transparent object reconstruction, following TransparentGS (Huang, 2025). Moreover, we consider the spectrum of a non-lambertian surface as a linear combination of the diffuse and specular components, and try to introduce the Dichromatic model in our framework for its reconstruction.

---

> ### Author Response · Authors · 2025-08-06
>
> Dear reviewer, thank you so much for your thoughtful feedback and support of our work. We have carefully provided point-by-point responses for each of your comments. If you have any further questions or concerns, please don’t hesitate to let us know. We are willing to provide additional theoretical analysis and experimental results to address your questions. Your support means a lot to us, and we truly appreciate your continued kindness and encouragement.

---

### Official Review · Reviewer_E8MT · 2025-07-05

**Clarity:** 3
**Significance:** 2
**Originality:** 2
**Rating:** 4
**Confidence:** 4

**Summary:**

This paper propose REArtGS, a method for reconstructing articulated objects from multi-view RGB images captured at two different states. The approach uses 3D Gaussian Splatting enhanced with unbiased-SDF regularization to generate high-quality surface meshes.

The key innovation is an "unbiased SDF guidance" that regularizes the Gaussian opacity fields, addressing insufficient 3D constraints in the reconstruction. The method learns kinematic parameters (translation and rotation) during optimization, enabling it to estimate the articulation model and generate intermediate poses between the two input states. Experiments on synthetic PartNet-Mobility and real-world AKB-48 datasets show REArtGS outperforms existing methods like PARIS.

**Questions:**

- **Inadequate  Evaluation**: Why are the critical kinematic parameter results (Ang Err, Pos Err) and part-level evaluations (CD-m, CD-s) relegated to the supplementary material when these metrics are fundamental for articulated object reconstruction? Additionally, why is ArtGS excluded from the kinematic parameter comparison in Table 7, given that it's a strong baseline for articulated object modeling? Moving kinematic evaluation to the main paper and including comprehensive comparison with ArtGS across all metrics would demonstrate the method's true performance against state-of-the-art approaches.

- **Questionable Technical Novelty Claims**: The "unbiased SDF regularization" is presented as a core contribution, but GOF already proposed using SDF regularization for 3D Gaussian learning. How does your approach fundamentally differ from GOF's method beyond applying it to articulated objects? What specific technical innovations justify this as a primary contribution rather than an engineering application?  Clear demonstration of novel technical components beyond existing SDF-Gaussian integration, or honest repositioning of the work as an engineering contribution with appropriate claims.

- **Single-Part Limitation Acknowledgment**: The method appears fundamentally restricted to objects with one movable part, yet this critical limitation is not clearly stated in the main paper. Honest acknowledgment of the single-part restriction in the main paper, with discussion of its implications and potential solutions, would improve the paper's transparency and help readers understand its scope.

**Ethical Concerns:**

["NO or VERY MINOR ethics concerns only"]

**Final Justification:**

Thanks to the author for the detailed rebuttal and supplementary experiments, which resolved most of my concerns, I have revised the final rating to 4

**Limitations:**

No acknowledgment that the method is fundamentally limited to objects with one movable part. Additionally, when the number of movable parts increases, reconstructing articulated objects from two states will be more challenging due to the occlusion between these parts.

**Quality:**

2

**Strengths And Weaknesses:**

- REArtGS demonstrates superior performance over existing methods on standard benchmarks, outperforming approaches like DITTO and PARIS on PartNet-Mobility and AKB-48 datasets.
- The method produces high-quality textured meshes and shows good real-world generalization, even in cluttered environments.
- Ablation studies validate the contribution of key components like SDF regularization.
# Weaknesses
- **Insufficient evaluation.** For articulated objects reconstruction, kinematic parameters are as the same significant as part meshes. However, I found that the authors put the evaluation results of kinematic parameters (Ang Err) and part meshes (CD-m) in the supplementary and ignore a strong baseline ArtGS. The authors need to explain the reason.
- **Insufficient Technical Innovation.** While the "unbiased SDF regularization" is highlighted as a core contribution, it is not an innovation proposed by the authors.  Using unbiased SDF regularization for 3D Gaussian learning has been proposed by GOF and this paper apply this for articulated objects reconstruction. I recognize the engineering contribution of this paper but it't not enough to be a core contribution.
- **Single-Part Restriction.** The method is limited to objects with only one movable part.

---

> ### Author Rebuttal · Authors · 2025-07-25
>
> We sincerely appreciate your valuable comments. The following is a point-by-point response to your questions. If you have any other questions, please let us know at the first time.  We earnestly look forward to the subsequent discussions with you.
>
> **1. Question on Insufficient Evaluation**
>
> Thank you for your insightful opinions. We also recognize the significance of joint parameters and part-level meshes for articulated objects. However, as the main topic of this paper is the surface reconstruction and generation of articulated objects, we moved the evaluation results of joint parameter and part-level mesh results to the Appendix taking the page limitation into consideration. Additionally, the surface generation results presented in the main draft are essentially high-dimensional mappings of part-level meshes. Realistic surface generation already indicates accurate part-level mesh extraction.
>
> As for not comparing the joint parameter and part-level mesh evaluation results of ArtGS, it is because ArtGS is essentially a method based on RGBD image inputs, while our method uses RGB images. We have already demonstrated in the main draft that our method outperforms ArtGS in surface reconstruction and generation when using the same RGB image inputs on both synthetic and real-world data. In kinematic parameter evaluation results of the supplementary material, we are more concerned with the comparison to PARIS, which also only uses RGB image inputs. Following your valuable suggestions, we have added  quantitative comparisons of joint parameter estimation and part-level mesh reconstruction between our method and ArtGS. To ensure a fair comparison, we train ArtGS using the same RGB image inputs as our approach. The results are presented below. The experimental results demonstrate that our method outperforms ArtGS in joint estimation and part-level mesh reconstruction.
>
> **Quantitative comparison of joint parameter estimation:**
> | Metrics       | Method         | Stapler |  USB  | Scissor | Fridge | Foldchair | Washer |  Oven  | Laptop | Blade  | Storage |  Mean  |
> |---------------|----------------|---------|-------|---------|--------|-----------|--------|--------|--------|--------|---------|--------|
> | Ang Err ↓     | ArtGS      | 0.062  | **0.034**  |  0.039 |  0.038 |   0.048  | 0.081  | 0.066 | 0.052  |  0.072 | **0.020**  | 0.051  |
> |               | REArtGS (Ours) | **0.042**   | 0.059 | **0.010**   | 0.006  | **0.013**     | **0.067**  | 0.031  | **0.012**  | 0.005  | 0.201   | **0.045**  |
> | Pos Err ↓       |ArtGS      | 0.011   | 0.002 |  0.001  | 0.003  |   0.000   | 0.017  |   0.004| **0.001**  | -      | -       | 0.004  |
> |               | REArtGS (Ours) | **0.002**   | **0.000** | **0.000**   | **0.000**  | 0.006     | **0.011**  | 0.004  | 0.003  | -      | -       | **0.003**  |
>
> **Quantitative comparison of part-level mesh reconstruction:**
> | Metrics       | Method         | Stapler |  USB  | Scissor | Fridge | Foldchair | Washer |  Blade  | Laptop | Oven  | Storage |  Mean  |
> |---------------|----------------|---------|-------|---------|--------|-----------|--------|--------|--------|--------|---------|--------|
> | CD-s ↓       | ArtGS      |  3.45 |  2.05 | 0.57  |  2.03 |   **0.16**  |  20.97 | **0.44** | 0.76 | 9.20  |  13.51 | 5.31  |
> |               | REArtGS (Ours) | **1.47**   | **0.82** | **0.25**   | **1.53**  | 0.31     | **13.04**  | 0.69  | **0.48**  | **6.59**  | **7.30**   | **3.25**  |
> | CD-m ↓    |  ArtGS      |   2.78 | 1.17 |  0.67  | 1.43  |    0.57  |  2.82 | **1.80**  | 0.99  |  2.12 | **7.18** | 2.15 |
> |               | REArtGS (Ours) | **1.18**   | **0.40** | **0.21**   | **0.66**  | **0.30**     | **0.42**  | 1.97  | **0.11**  | **0.25**    | 12.39    | **1.79**  |
>
>
> **2. Clarification on Technical Novelty**
>
> We sincerely appreciate your dedicated efforts in the research of the relevant works. To our best knowledge, our method first innovatively introduce the unbiased SDF regularization, establishing a connection between SDF and the Gaussian opacity field.
>
> **It is important to clarify that GOF is an opacity-based 3DGS method.** SDF is not incorporated throughout its entire  process and its mesh extraction solely relies on an opacity level set. Given that opacity cannot explicitly represent the relationship between 3D Gaussian primitives and the scene surface, the opacity-based method typically yields surface mesh results with artifacts distant from the surface.
>
> Besidse, integrating SDF into 3DGS poses a significant challenge. This is because 3DGS is fundamentally a discrete point-based rendering pipeline, while SDF is a continuous implicit function representation. Existing works, such as GSDF and 3DGSR, merely utilize SDF guidance for normal constraint and the pruning of Gaussian primitives and fall short of futher regularizing the opacity of 3D Gaussians, which serves as the core attribute in $\alpha$-blending.
>
> In contrast, our unbiased SDF regularization encourages the alignment of the Gaussian opacity field with SDF. Specifically, when the opacity reaches its maximum value, it drives the SDF value at the corresponding position to approach zero. This ensures that Gaussians near the surface exhibit high opacity values, eliminating artifacts away from the surface. Therefore, the unbiased SDF regularization effectively addresses the limitations of the opacity field in geometric representation and forms the core innovation of our method.
>
> Note the TSDF fusion method mentioned in GOF's paper is distinct from the neural SDF representation and is designed for   mesh extraction with discrete volume representations. Its SDF values are simply constructed through multi-view depth image fusion when extracting the mesh, rather than a continuous representation learned from the network.
>
> **3. Clarification on Single-Part Limitation**
>
> We sincerely appreciate your thorough consideration of our method. Our method is capable of effectively dealing with articulated objects exhibiting multiple movable parts. **We have already presented the high-quality reconstruction and generation results of an articulated object with three movable parts at the fourth row of Figure 6 in the main draft.** Moreover, we have elaborated in detail on the generation pipeline for articulated objects with multiple movable parts in Appendix C.4. We have put the details with italics below for your quick reference.
>
> *Mesh generation of multi-part articulated objects can be treated as a straightforward extension of the two-part task for our approach. We can generate meshes for multi-part objects in unseen states through a sequential learning strategy.  Specifically, given an articulated object with $k$ dynamic parts $\omega\_{d_k}$ , we start by fixing all dynamic parts $\omega\_{d\_{2}}, ..., \omega\_{d\_{k}}$ except $\omega\_{d\_1}$ . The object is then decomposed into two parts: a dynamic part $\omega\_{d\_1}$ and a static part $\omega_s$ (comprising all remaining points). After learning the motion and segmentation mask for $\omega\_{d\_1}$ , we sequentially process the next dynamic part $\omega\_{d\_2}$—with the static part updated as:  $\omega\_s = \hat{\omega}\_{d\_2} - \omega\_{d\_1}$ ,  where $\hat{\omega}\_{d\_2}$ denotes the remaining points excluding $\omega\_{d\_2}$ .  By parity of reasoning, we extend this process to learn the motions and segmentation masks for all $k$ dynamic parts. Finally, mesh generation for any unseen state of the multi-part object is achieved using the pipeline proposed in Sec. 3.2.*
>
>
> Regarding the occlusion among multiple parts, the two-stage multi-view image inputs can mutually complement their respective unseen regions. Specifically, since the Gaussian primitives in the reconstruction stage undergo further optimization in the generation stage, the images from each viewpoint in both stages can act as valid rendering supervisions for 3D Gaussian primitives. This allows our method to make full use of the multi-viewpoint images from the two stages to refine the Gaussian primitives. Please refer to the response to Reviewer BVdL for details. Consequently, in most scenes, the two-stage multi-view image inputs are adequate to effectively mitigate the occlusion problem among multiple parts.

---

> > ### Author Response · Authors · 2025-08-02
> >
> > Dear reviewer, we have provided point-by-point responses to your valuable comments. Please let us know if our responses have addressed your concerns so that we can fully resolve your issues before the discussion ends. Your feedback is of great importance to us.

---

> > ### Author Response · Authors · 2025-08-05
> >
> > Dear Reviewer, thank you sincerely for the time and effort in reviewing. As the discussion deadline approaches in just under three days, we would like to kindly ask whether you have any additional questions or if there are any further experiments you would like us to consider. We are truly happy to provide additional theoretical analysis or experimental results as soon as possible to help clarify any points of concern. Your feedback is truly valuable to our work REArtGS, and we would be grateful if we could receive your recognition.

---

> > ### Comment · Reviewer_E8MT · 2025-08-06
> >
> > Thanks to the author for the detailed rebuttal and supplementary experiments, which resolved most of my concerns, I have revised the final rating to 4

---

### Author Response · Authors · 2025-08-05
**Following up on our rebuttal and discussion**

Dear Area Chair and Reviewers, it has been about five days since the discussion period began. We have responded to all reviewer's comments and would like to again express our sincere gratitude for your time and insightful feedback. Hence, we summarize our main responses as follows:

1. Technical differences from existing papers, such as GOF, 3DGSR, NeuS (See point 2 of the response to Reviewer E8MT, point 4 of the response to Reviewer oBKh).
2. More quantitative results compared with ArtGS (See point 1 of the response to Reviewer E8MT).
3. Clarification of the reconstruction and generation for multi-part articulated objects (See point 3 of the response to Reviewer E8MT).
4. Input view setting and ablation of input view number (See point 2 of the response to Reviewer BVdL).
5. Explanation of $x_{i+1}$ (See point 1 of the response to Reviewer oBKh).
6. More results on general reconstruction datasets (See point 6 of the response to Reviewer Jnu3).
7. Comparison of training time (See point 3 of the response and official comment to Reviewer BVdL).

We hope these clarifications and additional experiments address your concerns. We are eager to engage in further discussion and are ready to answer any follow-up questions. We would be very grateful to hear your updated thoughts.

Sincerely,

Authors of the paper 14308.

---

### Note · Authors · 2025-08-11

We summarize the main clarifications during the rebuttal and discussion below.

**1. Originality of the unbiased SDF regularization**

Existing opacity-based 3DGS methods (GOF, 2DGS) typically yield surface mesh results with artifacts deviating from the surface, since the opacity cannot explicitly model the relationship between 3D Gaussians and scene surface. Although several works (GSDF, 3DGSR) introduce SDF to improve surface reconstruction quality, their improvements are limited to normal constraints and Gaussian pruning.

In contrast, this paper proposes an innovative integration of SDF in 3DGS.  Our rendering pipeline combines SDF with Gaussian's opacity, fusing global spatial information with local features of 3D Gaussians.  Moreover, **the proposed unbiased SDF regularization ensures that 3D Gaussians near the surface exhibit high opacity values, eliminating artifacts deviating from the surface.**

**2. Comparison of training time**

We have already provided the quantitative results of training time in the rebuttal and official comment to Reviewer BVdL. The experimental results prove that our method maintains competitive efficiency compared with existing methods. Note that we have implemented **engineering optimizations** in our approach to reduce the training time. **The engineering optimizations are simply code-level improvements without adjustment to our framework.** We also present surface reconstruction and generation results with the acceleration, which demonstrate slight affect and maintain SOTA performance.  We will add a brief description of the optimizations in the Implementation section of the final version.

**3. More experiments for addressing concerns**

Following the valuable suggestions of the reviewers, we have conducted more experiments to validate our effectiveness, such as more quantitative results compared with ArtGS and more results on general reconstruction datasets. **However,  the experiments in this paper are sufficient to prove that our method can achieve high-quality surface reconstruction and generation results of articulated objects.** Please refer to the rebuttal to Reviewer E8MT and Reviewer oBKh for details.

**4. Reconstruction and generation for multi-part articulated objects**

Our method is capable of effectively dealing with articulated objects exhibiting multiple movable parts. We have already presented the qualitative results in Fig. 6 in the main draft and the detail pipeline in Appendix C.4.

---

### Decision · Program_Chairs · 2025-09-17

**Decision:**

Accept (poster)

**Comment:**

This paper proposes a method for reconstructing articulated objects using 3D Gaussian Splatting from multi-view RGB images of two arbitrary articulation states.

On the positive side, all reviewers appreciated the overall engineering contribution of the work and the usefulness of the unbiased Signed Distance Field (SDF) guidance to regularize Gaussian opacity fields.

In terms of potential weaknesses, some concerns were raised, particularly regarding novelty and technical differences with GOF, 3DGSR, and NeuS. Reviewers also requested clarifications on the input view setting, training times, reconstruction of multi-part articulated objects, and results on general reconstruction datasets (e.g., Tanks and Temple, DTU). The authors provided the additional results and clarifications, which addressed these concerns.

In the end, all five reviewers leaned positive (three borderline accepts, two accepts). The AC concurs with this assessment and recommends acceptance. The authors are strongly encouraged to incorporate the additional results and clarifications from the rebuttal and discussion into the final version of the paper.